# ER retention is imposed by COPII protein sorting and attenuated by 4-phenylbutyrate

Wenfu Ma[1], Elena Goldberg[1], Jonathan Goldberg[1,2]*

[1]Structural Biology Program, Memorial Sloan Kettering Cancer Center, New York, United States; [2]Howard Hughes Medical Institute, New York, United States

**Abstract** Native cargo proteins exit the endoplasmic reticulum (ER) in COPII-coated vesicles, whereas resident and misfolded proteins are substantially excluded from vesicles by a retention mechanism that remains unresolved. We probed the ER retention process using the proteostasis regulator 4-phenylbutyrate (4-PBA), which we show targets COPII protein to reduce the stringency of retention. 4-PBA competes with p24 proteins to bind COPII. When p24 protein uptake is blocked, COPII vesicles package resident proteins and an ER-trapped mutant LDL receptor. We further show that 4-PBA triggers the secretion of a KDEL-tagged luminal resident, implying that a compromised retention mechanism causes saturation of the KDEL retrieval system. The results indicate that stringent ER retention requires the COPII coat machinery to actively sort biosynthetic cargo from diffusible misfolded and resident ER proteins.

## Introduction

Resident chaperones and bound folding intermediates make up the majority of ER content and must be retained at that site (*Barlowe and Helenius, 2016*; *Gomez-Navarro and Miller, 2016*). A simple model for retention would be for resident proteins to consolidate into immobile networks too large to enter COPII vesicles, but data indicate instead that resident complexes diffuse relatively freely in the ER (*Hendershot, 2000*; *Nehls et al., 2000*), leaving the mechanism of ER retention an open problem. It is known that residents can escape the ER and are selectively retrieved from post-ER compartments, in particular by the KDEL receptor; however, when the KDEL signal is removed from a resident chaperone, its exit is considerably slower than the bulk flow rate (*Barlowe and Helenius, 2016*; *Munro and Pelham, 1987*). We sought to test the hypothesis that COPII-associated machinery restricts the passive uptake into vesicles of mobile resident complexes and associated misfolded proteins.

Candidate machinery for retention includes the p24-family proteins. These highly conserved transmembrane proteins cycle continuously between ER and Golgi membranes as abundant constituents of COPI and COPII vesicles (*Kaiser, 2000*; *Otte et al., 2001*; *Stamnes et al., 1995*). Mutation of p24 genes results in post-ER trafficking of a misfolded protein in Caenorhabditis elegans (*Wen and Greenwald, 1999*). Also, deletion of p24 genes in yeast causes misfolded proteins and chaperones to escape the ER, with the latter being secreted into the extracellular medium; however, these effects may be attributable in part to a chronic unfolded protein response (UPR) in the knockout strains (*Belden and Barlowe, 2001*; *Copic et al., 2009*; *Elrod-Erickson and Kaiser, 1996*). In a distinct line of research, the small molecule 4-phenylbutyrate has been shown to restore trafficking of various misfolded, ER-trapped mutant proteins expressed in cultured cells, including for example the △F508 mutant cystic fibrosis transmembrane conductance regulator (*Liu et al., 2004*; *Rubenstein et al., 1997*; *Zode et al., 2012*). Generally, 4-PBA also reduces ER stress and attenuates

*For correspondence: goldberj@mskcc.org

the UPR in these systems. The target and mechanism of action of 4-PBA are unknown. Its efficacy in promoting trafficking of a broad range of misfolded proteins, including transmembrane and luminal secretory molecules, rules out function as a pharmacologic chaperone that binds and stabilizes the folded state of mutant protein in the ER. Instead, 4-PBA has been referred to as a proteostasis regulator to convey that it can restore ER and trafficking homeostasis (*Balch et al., 2008*). We theorized that the ability of 4-PBA to restore trafficking of misfolded proteins in cultured cells phenocopies p24 genetic knockout in yeast and worms. Notably, 4-PBA has a structural formula that resembles the cytoplasmic ER export signal of p24 proteins—one or two C-terminal hydrophobic residues, termed the ΦC motif (*Nakamura et al., 1998*; *Nufer et al., 2002*). On this basis, we evaluated COPII protein as the mechanism-of-action target of 4-PBA, and then utilized 4-PBA to probe the ER retention process.

## Results and discussion

### Binding site on COPII protein for ER export signals containing the ΦC motif

The packaging of cargo transmembrane proteins into COPII-coated vesicles entails a direct interaction between COPII protein and cytoplasmically-oriented ER export signals. About a half dozen export signals have been identified to date, almost all comprising short peptide motifs, all of them binding to the Sec24 subunit of the COPII inner coat (*Gomez-Navarro and Miller, 2016*). The ΦC export signal was identified in studies of p24-family and ERGIC-53 proteins, and comprises a C-terminal signature of a pair of hydrophobic residues or a single terminal valine residue (*Nakamura et al., 1998*; *Nufer et al., 2002*). Despite clear evidence for the ΦC motif as a signal for ER export, its binding site on the COPII coat has been difficult to identify biochemically. We reasoned that the shortness of the cytoplasmic tail (10–15 cytoplasmic residues) of p24 and ERGIC-53 proteins contributes significantly to tight binding of the ΦC signal to membrane-associated COPII protein (because the encounter will be appreciably restricted to two dimensions) and that this affinity enhancement would be lost in a solution-based biochemical search for the interaction. To resolve this problem we used protein crystallography to identify the binding interaction directly. A series of ΦC-signal peptides was synthesized, based on a core sequence of yeast Emp24p, and individual peptides were soaked at high concentrations into two crystal forms of human COPII protein: one comprising Sec23a/Sec24a•Sec22b, the other Sec24c (*Mancias and Goldberg, 2008*). X-ray diffraction data were collected from cryopreserved crystals (see Materials and methods). For all ΦC sequences, we observed clear evidence for peptide binding to a site on Sec24a in the Sec23a/Sec24a•Sec22b crystals, based on the presence of residual electron density that matches the peptide sequence (this binding site is occluded in the Sec24c crystals). *Figure 1A–E* show four of the COPII•ΦC crystal structures (additional structures are presented in *Figure 1—figure supplement 1*; X-ray crystallography statistics and ΦC peptide sequences are reported in *Table 1*).

The ΦC motif contacts a site on Sec24a that is also the binding site for the LxxLE- and DxE-class export motifs (*Miller et al., 2003*; *Mossessova et al., 2003*; see *Figure 1—figure supplement 1*). Referred to as the B site, this was identified previously as a potential binding site for p24 proteins, as mutation of residues therein impairs packaging of p24 proteins into COPII vesicles budded from yeast microsomal membranes; however, the conclusion was a tentative one since residual p24 packaging was observed (*Miller et al., 2003*). The finding that multiple classes of export motifs bind to a common site is surprising, but the COPII•ΦC crystal structures highlight that the common molecular feature is carboxylate-group recognition by the clustered arginine residues of the B site—the terminal carboxylate group of the ΦC motif adopts a similar position and bonding arrangement to the carboxylate of the glutamate side chains of LxxLE and DxE motifs (*Figure 1—figure supplement 1*; *Mancias and Goldberg, 2008*; *Mossessova et al., 2003*). Thus, specific recognition of the ΦC motif involves the terminal carboxylate group bonding to Arg750 and Arg752 of Sec24a, with the penultimate hydrophobic residue of the motif nestling against Tyr496, and the terminal hydrophobic residue fitting to a pocket of hydrophobic side chains (*Figure 1B–E* and S1). These molecular features are in concordance with functional data concerning ΦC motifs as ER export signals (*Nakamura et al., 1998*; *Nufer et al., 2002*). Evidently, the B-site hydrophobic pocket accommodates a range of hydrophobic side chains; however, a terminal valine is the most favored residue

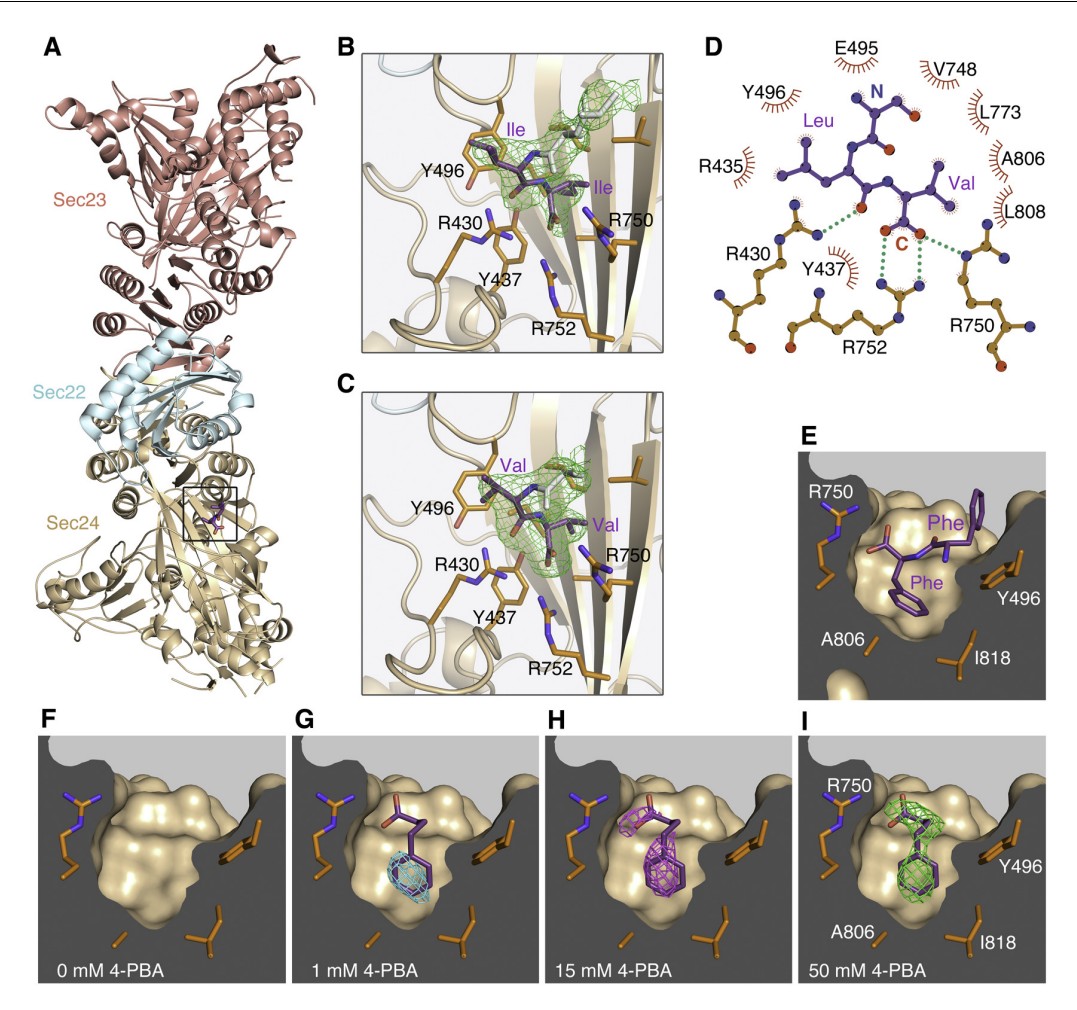

**Figure 1.** COPII binding site for the ΦC signal motif and 4-PBA located by X-ray crystallography. (**A**) Crystal structure of the COPII•SNARE complex comprising human Sec23a/Sec24a•Sec22b bound to a ΦC peptide (boxed). The concave, membrane-binding surface of the coat faces right. (**B**) Close-up view of the B site on Sec24a with bound ΦC sequence containing the terminal Ile-Ile motif (colored purple) of the S. cerevisiae p24 protein Erv25p. Green contour lines show residual electron density ($F_o$-$F_c$ synthesis with no phase bias) at 2.6 Å resolution, contoured at 2.6 σ. Electron density is also observable for N-terminal residues of the peptide (residues colored light grey). Key side chains of Sec24 are labeled (colored orange). See *Table 1* for details of peptide-bound crystal structures. (**C**) Residual ($F_o$-$F_c$) electron density (2.6 Å resolution, 2.6 σ contour) for a bound ΦC peptide containing the terminal Val-Val motif of human p24β1. (**D**) Schematic diagram shows the bonding arrangement to the potent ΦC motif Leu-Val. Diagram is based on the structure of Sec23a/Sec24a•Sec22b bound to the sequence EVTSLV (see *Table 1*). The label N denotes the serine residue upstream of the Leu-Val motif. (**E**) Conformation of a ΦC peptide containing the terminal Phe-Phe motif of ERGIC-53 bound to Sec24a (drawn as a surface representation). (**F–I**) Residual ($F_o$-$F_c$) electron density (2.8 Å resolution, 2.3 σ contour, no phase bias) at the B site for crystals soaked in 4-PBA at the indicated concentrations. No electron density is observable for ordered water or solute molecules at this contour level in the 0 mM 4-PBA experiment (**F**). Note how 4-PBA (colored purple) mimics the terminal phenylalanine side chain and carboxylate group in (**E**). To aid comparisons, the electron density calculations are all truncated at 2.8 Å; see *Table 2* for details of X-ray datasets. See also *Figure 1—figure supplement 1*, and *Tables 1* and *2*.

The following figure supplement is available for figure 1:

**Figure supplement 1.** Comparison of the COPII binding site occupied by ΦC, DxE and LxxLE signal sequences.

because its fit to the pocket allows optimal geometry for bonding to the terminal carboxylate group (shown schematically in *Figure 1D*). A subtle change of the bonding geometry is observed for the Phe-Phe motif of ERGIC-53, in which the larger terminal side chain of phenylalanine is accommodated by a shift of the carboxylate group to bond Arg750 in a bidentate fashion (*Figure 1E*). This detail has relevance to the binding mode of 4-PBA, as described below.

**Table 1.** Data Collection and Refinement Statistics – ΦC signal complexes.

| COPII protein complex | Sec23/24•22 | Sec23/24•22 | Sec23/24•22 | Sec23/24•22 | Sec23/24•22 | Sec23/24•22 | Sec23/24•22 |
|---|---|---|---|---|---|---|---|
| Peptide | EVTSLV | EVTSVV | EVTSII | EVTSSV | EVTSFA | EVTSFF | EVTSLL |
| Protein Data Bank ID | 5VNE | 5VNF | 5VNG | 5VNH | 5VNI | 5VNJ | 5VNK |
| Space group: | C2 | C2 | C2 | C2 | C2 | C2 | C2 |
| copies/asymmetric unit | 1 | 1 | 1 | 1 | 1 | 1 | 1 |
| Cell parameters a, b, c (Å) | 148.4, 96.8, 127.0 | 147.8, 97.1, 128.8 | 148.0, 96.9, 130.0 | 147.9, 96.9, 127.3 | 147.6, 96.1, 130.0 | 147.9, 96.9, 129.5 | 148.2, 97.6, 130.5 |
| Cell parameters $\beta$ (°) | 91.7 | 90.1 | 90.1 | 91.6 | 90.1 | 90.2 | 90.1 |
| Data processing | | | | | | | |
| Resolution (Å) | 50–2.7 | 50–2.6 | 50–2.6 | 50–2.6 | 50–2.8 | 50–2.8 | 50–2.6 |
| Rmerge (%)[†] | 8.9 (70.2) | 9.6 (52.0) | 11.4 (55.0) | 8.0 (77.4) | 11.6 (58.2) | 11.6 (58.2) | 12.6 (78.9) |
| I/σ | 24.5 (2.6) | 18.0 (1.5) | 16.2 (1.1) | 29.5 (2.5) | 12.6 (1.0) | 12.6 (1.0) | 30.1 (2.3) |
| Completeness (%) | 99.8 (100) | 99.0 (99.5) | 98.8 (99.2) | 99.5 (100) | 98.2 (99.5) | 98.2 (99.5) | 99.5 (99.0) |
| Redundancy | 4.0 (4.0) | 3.7 (3.1) | 3.2 (2.4) | 4.2 (4.2) | 2.5 (2.3) | 2.5 (2.3) | 3.8 (3.7) |
| Refinement statistics | | | | | | | |
| Data range (Å) | 50–2.7 | 50–2.6 | 50–2.6 | 50–2.6 | 50–2.8 | 50–2.8 | 50–2.6 |
| Reflections | 52341 | 54635 | 54087 | 61690 | 40031 | 44706 | 60273 |
| Nonhydrogen atoms | 12597 | 12619 | 12558 | 12577 | 12545 | 12535 | 12559 |
| Water molecules | 99 | 111 | 59 | 81 | 46 | 77 | 60 |
| R.m.s. Δ bonds (Å)[‡] | 0.6 | 0.6 | 0.7 | 0.6 | 0.6 | 0.5 | 0.6 |
| R.m.s. Δ angles (°)[‡] | 0.002 | 0.003 | 0.002 | 0.002 | 0.003 | 0.002 | 0.002 |
| R-factor (%)[§] | 22.2 | 20.5 | 20.7 | 21.5 | 21.7 | 21.3 | 19.6 |
| Rfree (%)[§, #] | 26.8 | 24.1 | 25.7 | 25.3 | 27.3 | 26.0 | 25.1 |

[*]Highest resolution shell is shown in parenthesis.

[†]$R_{merge} = 100 \times \sum_h \sum_i | I_i(h) - <I(h)> | / \sum_h <I(h)>$, where $I_i(h)$ is the $i$th measurement and $<I(h)>$ is the weighted mean of all measurement of $I(h)$ for Miller indices h.

[‡]Root-mean-squared deviation (r.m.s. Δ) from target geometries.

[§]R-factor = $100 \times \sum |F_P - F_{P(calc)}| / \sum F_P$.

[#]$R_{free}$ was calculated with 5% of the data.

## 4-PBA mimics the ΦC signal motif to bind the COPII coat

As an initial test of the hypothesis that 4-PBA targets the COPII coat, crystals of Sec23a/Sec24a•Sec22b were equilibrated with solutions containing 4-PBA at concentrations of 1, 15 and 50 mM (see Materials and methods and *Table 2*). (We note that the numerous reports of efficacy in cell culture models of ER stress and cargo mistrafficking utilize 4-PBA in the 5–20 mM range.) Difference Fourier maps calculated from X-ray diffraction data show clear evidence for binding of 4-PBA to the B site of Sec24a (*Figure 1F–I*). 4-PBA closely mimics the terminal phenylalanine side chain and carboxylate group of the Phe-Phe motif of ERGIC-53 (compare *Figure 1E and I*). The residues contacting 4-PBA are almost invariant among the four human Sec24 paralogs; only Sec24a residue I818 (drawn in *Figure 1I*) varies, and the change is a conservative one to leucine in Sec24c/d.

In summary, the crystallographic result gives evidence that 4-PBA binds the COPII coat at relevant concentrations of the small molecule.

## Vesicle packaging of p24 proteins is inhibited by 4-PBA and by a ΦC binding-site mutation

Next we tested the proposition that p24 proteins are captured into vesicles by the B site of COPII and that this site is targeted by 4-PBA. For this purpose we used a mammalian permeabilized-cell assay that measures the selective packaging of cargo proteins into COPII-coated vesicles (see

**Table 2.** Data collection and refinement statistics – 4-PBA crystallography.

| COPII protein complex | Sec23/24•22 | Sec23/24•22 | Sec23/24•22 | Sec23/24•22 |
|---|---|---|---|---|
| 4-PBA Concentration | 1 mM | 15 mM | 50 mM | 0 mM |
| Protein Data Bank ID | 5VNL | 5VNM | 5VNN | 5VNO |
| Space group | C2 | C2 | C2 | C2 |
| Copies/asymmetric unit | 1 | 1 | 1 | 1 |
| Cell parameters: a, b, c (Å) | 147.8, 96.8, 129.6 | 149.1, 96.8, 130.6 | 148.02, 97.7, 128.9 | 148.4, 96.9, 130.5 |
| cell parameters: $\beta$ (°) | 90.2 | 90.2 | 89.6 | 89.6 |
| Data processing | | | | |
| Resolution (Å) | 50–2.4 | 50–2.8 | 50–2.5 | 50–2.9 |
| Rmerge (%)[†] | 8.9 (45.9) | 9.9 (80.9) | 9.3 (56.7) | 7.6 (86.2) |
| I/$\sigma$ | 16.1 (1.2) | 11.6 (1.2) | 13.3 (1.3) | 15.9 (1.7) |
| Completeness (%) | 98.9 (95.9) | 99.3 (99.5) | 99.2 (99.1) | 98.6 (99.5) |
| Redundancy | 2.9 (1.8) | 3.7 (3.9) | 3.0 (2.3) | 3.6 (3.8) |
| Refinement statistics | | | | |
| Data range (Å) | 50–2.4 | 50–2.8 | 50–2.5 | 50–2.9 |
| Reflections | 61011 | 43047 | 53119 | 39048 |
| Nonhydrogen atoms | 12471 | 12462 | 12491 | 12477 |
| Water molecules | 50 | 45 | 59 | 48 |
| R.m.s. Δ bonds (Å)[‡] | 0.6 | 0.6 | 0.6 | 0.6 |
| R.m.s. Δ angles (°)[‡] | 0.003 | 0.002 | 0.002 | 0.003 |
| R-factor (%)[§] | 21.1 | 21.3 | 21.2 | 20.9 |
| Rfree (%)[§, #] | 24.6 | 26.0 | 25.1 | 25.3 |

[*]Highest resolution shell is shown in parenthesis.

[†]$R_{merge} = 100 \times \sum_h \sum_i | I_i(h) - <I(h)> | / \sum_h <I(h)>$, where $I_i(h)$ is the $i$th measurement and $<I(h)>$ is the weighted mean of all measurement of $I(h)$ for Miller indices h.

[‡]Root-mean-squared deviation (r.m.s. Δ) from target geometries.

[§]R-factor = $100 \times \sum |F_P - F_{P(calc)}| / \sum F_P$.

[#]$R_{free}$ was calculated with 5% of the data.

Materials and methods). Vesicle budding is supported in the assay by purified recombinant COPII coat proteins (Sec23/24, Sec13/31 and Sar1), or mutants thereof, in the absence of cytosol (*Adolf et al., 2016*; *Kim et al., 2007*; *Mancias and Goldberg, 2008*).

We measured the packaging of a series of cargo proteins into vesicles budded from permeabilized CHO (T-Rex-CHO-K1) cells (*Figure 2*). Cargo packaging was efficient (*Table 3*) and COPII-dependent, and the translocon accessory protein Ribophorin I was absent from all vesicle fractions, indicating that ER membrane integrity was preserved. To test the role of the B site in packaging p24 and ERGIC-53 proteins, we introduced the mutation L750W into Sec24d. We have modeled the Sec24d-L750W mutation in *Figure 2B* to indicate that the tryptophan side chain is restricted to a single rotamer, and this conformation is predicted to block binding of all ligands to the B site. This mutation was designed previously for a study of yeast COPII (*Miller et al., 2003*), where it was reported that the corresponding Sec23/Sec24-L616W mutant protein lost all ability to package LxxLE- and DxE-signal cargoes, and exhibited reduced packaging efficiency of Erp1p (a p24 protein) and Emp47p (yeast ERGIC-53 homolog); COPII-coated vesicles budded with the mutant protein exhibited negligible differences in size and morphology when compared to wild-type vesicles (*Miller et al., 2003*). As seen in *Figure 2A and C*, the uptake of p24 proteins (p24α3 and p24β1) and ERGIC-53 into vesicles by Sec23a/24d-L750W mutant COPII was substantially reduced, to just 5–8% of that obtained with wild-type COPII. However, the total yield of COPII vesicles is significantly lower for the mutant coat, which we infer is due to the decreased number of contacts made between

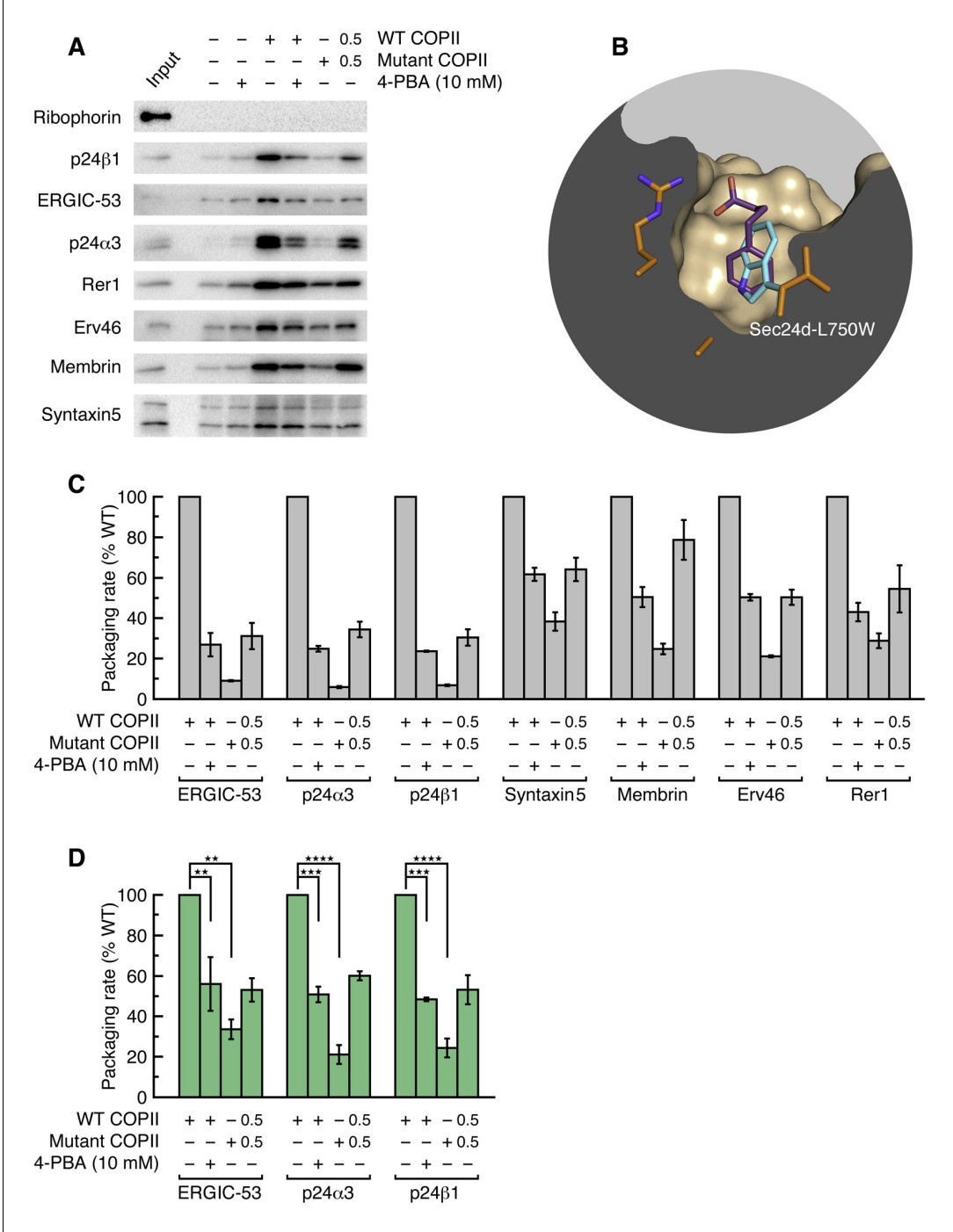

**Figure 2.** Application of 4-PBA or mutation of the B site of COPII reduces the packaging of p24 proteins in vesicles. (**A**) Reconstituted COPII budding reactions were performed using permeabilized CHO (T-Rex-CHO-K1) cells, using either wild-type or mutant (Sec24d-L750W) COPII protein. Protein contents of the isolated vesicles were analyzed by immunoblotting. Lane labeled Input represents 1.1% of the starting permeabilized cell membranes used in a budding reaction; all other lanes represent the vesicle product of 100% of the starting membranes. The syntaxin5 antibody recognizes both the short and long forms of the SNARE protein. These are representative blots from experiments performed in triplicate. (**B**) Close-up view of the B site of Sec24d, oriented as in *Figure 1E–I*, showing 4-PBA (colored purple) and a modeled position of the tryptophan side chain (cyan) in the Sec24d-L750W mutant protein used in budding reactions. (**C**) The packaging rate for cargo proteins was determined by immunoblotting as in (**A**) and densitometry of chemiluminescent signals. For each cargo molecule, data are normalized to be a percentage of the packaging rate measured using wild-type COPII protein. Bar

*Figure 2 continued on next page*

*Figure 2 continued*

graphs show mean ± s.e.m. (n = 3). (D) Data in (C) were corrected for treatment-dependent (4-PBA or mutant COPII) changes in budding yield (using the mean values of the packaging rates for Syntaxin 5, membrin, Erv46 and Rer1). For presentation clarity, data are normalized as in (C). Statistical analysis was performed on data prior to normalization in order to preserve statistical distributions (n = 3; ANOVA test, Bonferroni-Holm post hoc, **p<0.01, ***p<0.001 and ****p<0.0001). Bar graphs show mean ± s.e.m. See also *Table 3*.

coat protein and transmembrane cargo, a phenomenon reported previously for mammalian and yeast COPI-coated vesicles (*Bremser et al., 1999*; *Sandmann et al., 2003*). To control for the change in budding yield, we measured the packaging rates of four proteins that do not bind the B site (*Figure 2A*): the recycling receptors Erv46 and Rer1, and the SNARE proteins Syntaxin5 and Membrin (GS27) that utilize a distinct IxM motif for binding Sec24d (*Adolf et al., 2016*; *Mancias and Goldberg, 2008*). The uptake into vesicles is highly correlated for these four cargoes (*Figure 2C*), and the packaging rates for p24 proteins were corrected on this basis (see Quantitation and Statistical Analysis). Thus, blockade of the B site in Sec24d-L750W reduced the packaging rate of p24α3 and p24β1 to 20–25% of the wild-type level (*Figure 2D*). This residual p24 packaging is at a similar level to that reported for yeast B-site-mutant COPII (*Miller et al., 2003*), and we infer that it constitutes the bulk flow (passive) level of uptake of these proteins.

Significantly, when COPII vesicles were budded in the presence of 10 mM 4-PBA, we observed a 50% reduction in the packaging rate of p24 proteins and ERGIC-53 (*Figure 2A and D*). This concentration of 4-PBA is slightly above its $EC_{50}$ of 6.4 mM as measured in a cell-based transport assay (see *Figure 3B* and below); thus, a 50% packaging rate is an expected degree of inhibition given that complete blockade of the B site in the COPII mutant reduces packaging to 20–25% of wild-type level. Finally, we assessed whether the effects of 10 mM 4-PBA on cargo packaging are consistent with a partial occupancy of the B site, as emulated by a 50/50 mixture of wild-type and mutant COPII proteins. *Figure 2A and C* show that the effects of the two treatments on the seven cargo proteins are qualitatively and quantitatively highly similar. Together with the crystallographic observations,

**Table 3.** Cargo packaging efficiency measured for wild-type COPII protein.

| | Packaging efficiency (Amount in vesicle as % of protein present in starting permeabilized cells) | |
| --- | --- | --- |
| | CHO-LDLR$_{G544V}$ cells | Wild-Type CHO cells |
| **Cargo** | | |
| ERGIC-53 | 4.5 | – |
| p24δ1 | 7.7 | 4.6 |
| p24α2 | 8.7 | 10.9 |
| p24β1 | 10.6 | – |
| p24α3 | 6.4 | 6.4 |
| Syntaxin 5 | 0.89 | 0.96 |
| Membrin | 4.8 | – |
| Erv46 | 6.6 | 3.9 |
| Rer1 | 11.5 | 5.4 |
| **ER resident** | | |
| Calnexin | 0.32 | 0.065 |
| ERp57 | 0.63 | 0.12 |
| BiP | 0.87 | – |
| GRP94 | 0.25 | 0.07 |

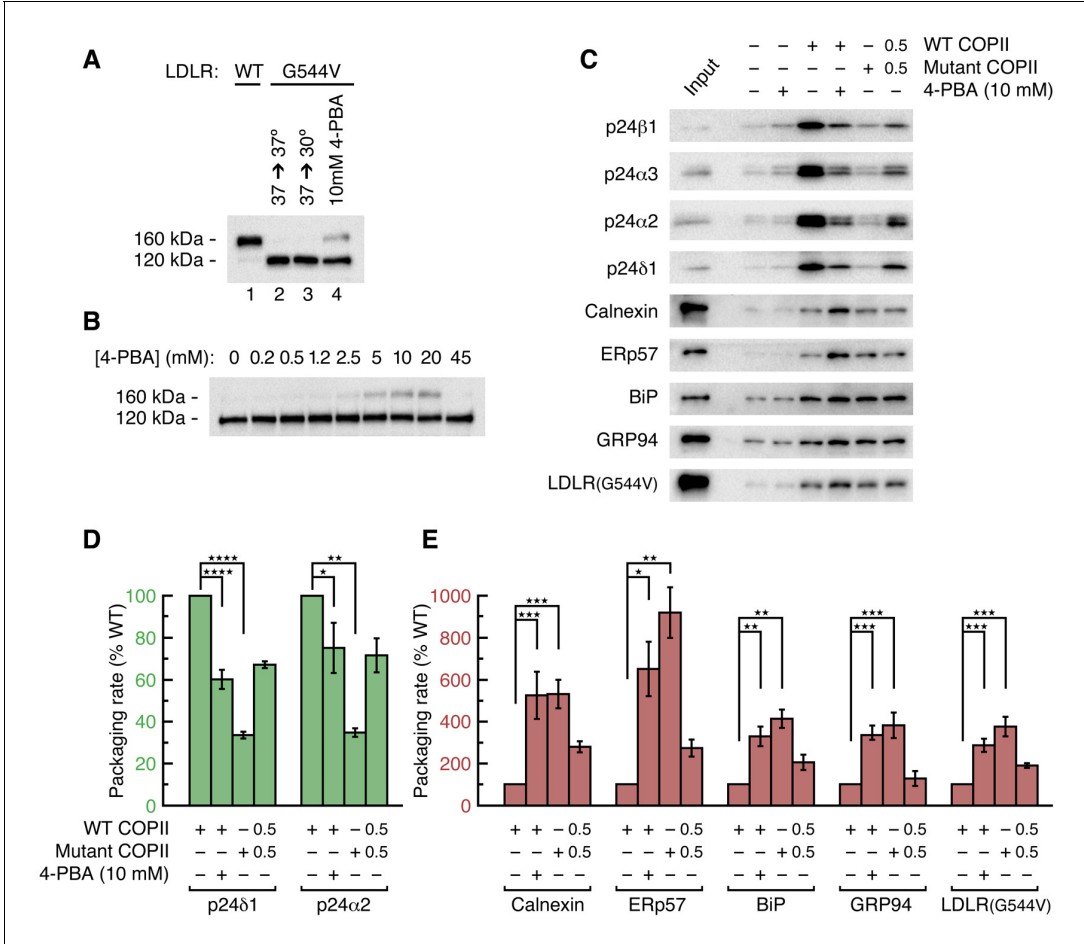

**Figure 3.** 4-PBA promotes packaging of ER resident proteins and mutant LDL receptor into COPII vesicles in vitro. (**A**) The G544V-mutant LDL receptor fails to reach the Golgi complex (*Tveten et al., 2007*). CHO cells stably transfected with wild-type (lane 1) or mutant LDL receptor were induced with tetracycline for 24 hr at 37°C before further incubation for 2 hr at 37°C (lane 2) or 30°C (lane 3), or 2 hr incubation at 37°C with 10 mM 4-PBA (lane 4). Cell lysates were analyzed by immunoblotting; labeling indicates the 160 kDa mature glycosylated and 120 kDa precursor forms of LDL receptor. (**B**) Dose-dependent restoration of trafficking of mutant LDL receptor by 4-PBA (see also *Tveten et al., 2007*). CHO cells expressing mutant LDL receptor were incubated with indicated concentrations of 4-PBA for 2 hr at 37°C. Cell lysates were analyzed as in (**A**). (**C**) The packaging of a series of p24 proteins, ER residents and mutant LDL receptor were assayed in scaled-up COPII budding reactions. Lane labeled Input represents 1.1% of the starting membranes; all other lanes represent the vesicle product of 100% of the starting membranes. Experiments were performed in triplicate. (**D**) Packaging rates for the p24-family proteins p24δ1 and p24α2, presented as in 2D (controlled by Syntaxin 5, membrin, Erv46 and Rer1 packaging rates). The data for p24α3 and p24β1 are in *Figure 2D* (n = 3; ANOVA test, Bonferroni-Holm post hoc, *p<0.05, **p<0.01 and ****p<0.0001). Bar graphs show mean ± s.e.m. (**E**) Graph shows packaging rates for four ER resident proteins and mutant LDL receptor. Note the change of scale of the y-axis (n = 3; ANOVA test, Bonferroni-Holm post hoc, *p<0.05, **p<0.01 and ***p<0.001). Bar graphs show mean ± s.e.m.

the results establish that the packaging of p24 proteins into COPII vesicles involves an interaction of the ΦC motif with the B site of Sec24, and that 4-PBA competes directly with p24 proteins to bind this site. Previously we showed that all four human paralogs of Sec24 package ERGIC-53 (*Mancias and Goldberg, 2008*). We now show that the ΦC motif and 4-PBA molecule bind to Sec24a (by crystallographic analysis) and to Sec24d (budding experiments); thus we surmise that COPII vesicles package p24 proteins to high levels (*Otte et al., 2001*) because all paralogs of Sec24 can bind the ΦC motif, and that all are targets of 4-PBA.

Previous studies established that the LxxLE and DxE signals bind only to the B site of Sec24a and Sec24b paralogs, and structural analysis indicated that Sec24c and Sec24d fail to bind these signals due to subtle residue changes in the vicinity of the B site (*Mancias and Goldberg, 2008*). Specifically, residue Leu773 (Sec24a numbering), which neighbors the key B-site residue Arg750 (see

*Figure 1—figure supplement 1*), is changed to aspartic acid in Sec24c and Sec24d. Whereas the LxxLE and DxE signals are predicted to be sensitive to this change, it is evident from the current structural and functional data that the ΦC signal and 4-PBA are indifferent to the change.

## 4-PBA promotes COPII packaging of resident proteins and ER-trapped mutant LDL receptor

We assessed whether the depletion of p24 proteins from COPII vesicles by 4-PBA and by coat mutation would produce a defect in ER retention and cause packaging of resident and misfolded proteins. We began by measuring the packaging of resident proteins and a misfolded mutant LDL receptor into COPII vesicles budded from permeabilized cells (*Figure 3*). An early application of the permeabilized-cell system indicated that it can faithfully reconstitute ER retention, as export of the thermoreversible variant of vesicular stomatitis virus (VSV) G protein from the ER was shown to be temperature dependent (*Beckers et al., 1987*). VSV G protein is not an optimal model for our analysis since it is not constitutively misfolded and because it employs a DxE signal for potent uptake into COPII vesicles via the B site of Sec24a/b paralogs (*Mancias and Goldberg, 2008*; *Nishimura et al., 1999*). Instead, we used a well-characterized system comprising a CHO (T-Rex-CHO-K1) cell clone expressing the G544V-mutant human LDL receptor under a tetracycline-inducible promoter (*Tveten et al., 2007*). This is a naturally occurring transport-defective (class 2) mutant LDL receptor with a point mutation in the luminal β-propeller domain that causes ER trapping with chaperone association (*Sørensen et al., 2006*). Importantly, 4-PBA restores its transport to the plasma membrane (*Tveten et al., 2007*). We verified that the mutant LDL receptor is not thermoreversible, since there was no evidence for transport to the Golgi complex during a 2 hr temperature shift of the cells to 30°C (*Figure 3A*; transport to the Golgi is monitored by the acquisition of Golgi-specific glycosylation to yield a 160 kDa form). And we observed that application of 10 mM 4-PBA to cells restored the trafficking of as much as 25% of the receptor (*Figure 3B*), in line with the results reported in *Tveten et al. (2007)*.

*Figure 3C* shows the results of a COPII budding experiment in which we probed for the packaging of a series of p24 proteins, ER residents and G544V-mutant LDL receptor. Remarkably, the depletion from vesicles of p24 proteins caused by 10 mM 4-PBA and by B-site mutation is accompanied by a significant enrichment of residents and mutant LDL receptor (*Figure 3C*; *Figure 3D and E* show the packaging rates corrected for change in budding yield). Vesicles budded with B-site-mutant COPII package five times as much calnexin as wild-type vesicles; the enrichment factors for BiP, GRP94 and mutant LDL receptor are all approximately 4-fold, and the protein disulfide isomerase ERp57 (PDIA3) is enriched 9-fold (*Figure 3E*). Measurements taken from the same vesicle pools for four p24 proteins show on average 3.5-fold depletion (*Figures 2D* and *3D*). Thus, protein sorting by the COPII coat yields a 15- to 25-fold purification of p24 proteins from ER residents—purification here refers to the product of resident-protein depletion and p24 enrichment in wild-type vesicles relative to B-site-mutant vesicles. Note that the degree of purification is inherent to the raw data in *Figure 3C*; i.e. it is unaffected by the corrections for changes in budding yield that are made for data presentation in *Figures 2D*, *3D and E*.

A corollary of these observations is that resident-protein complexes are sufficiently small to be packaged into mutant and 4-PBA-treated COPII vesicles. Thus, the reciprocal relationship that we observe between packaging of p24 proteins and residents is consistent with the model of an active sorting mechanism involving p24-dependent exclusion of diffusible resident-protein complexes.

## Protein sorting by COPII vesicles reconstituted using unstressed cells

The COPII sorting process indicated by our data could be a core feature of the secretory pathway or it could operate upon breakdown of normal retention mechanisms triggered by the accumulation of misfolded protein in the ER and the ensuing UPR. Specifically, the expression of ER-trapped G544V-mutant LDL receptor in the CHO cell line has been shown to induce ER stress and UPR activation, such that downstream markers of stress, including ER chaperone levels, are maximally expressed at the time of cell harvesting for our budding experiments (*Sørensen et al., 2006*). Understanding of the effect of the UPR on ER retention is incomplete. An early study showed that a fraction of the thermoreversible VSV G protein exits the ER following prolonged incubation at the restrictive temperature, is accompanied by BiP, and recycles to the ER via the cis-Golgi and intermediate

compartment (*Hammond and Helenius, 1994*). Further instances of ER-to-Golgi trafficking of mis-folded proteins in stressed mammalian and yeast cells have been reported (*Kincaid and Cooper, 2007*; *Vashist et al., 2001*; *Yamamoto et al., 2001*). An additional complication is that the deletion of p24 genes in yeast causes a chronic UPR (*Belden and Barlowe, 2001*), raising the possibility that the leakage phenotype of the yeast knockout is due entirely to UPR activation rather than a break-down of the active sorting mechanism that we posit.

To investigate further, we tested whether the COPII coat is capable of sorting resident proteins from cargo in the context of unstressed cells. As shown in *Figure 4*, the packaging of proteins into vesicles budded from parent CHO cells exhibits the same pattern of p24-dependence as from stressed cells. (Since the vesicle yield is reduced when budding from unstressed cells, we probed for a subset of cargo using only the highest quality antibodies available to us). Vesicles budded with B-site-mutant COPII are depleted of p24 proteins by a factor of ~8 fold (*Figure 4B*), yet the same pool of vesicles packages on average 5.5-fold higher levels of resident proteins (*Figure 4C*). This indicates a ~40 fold purification achieved by the COPII coat, though we note that this calculation is rather inaccurate because the packaging of residents by wild-type COPII is close to background

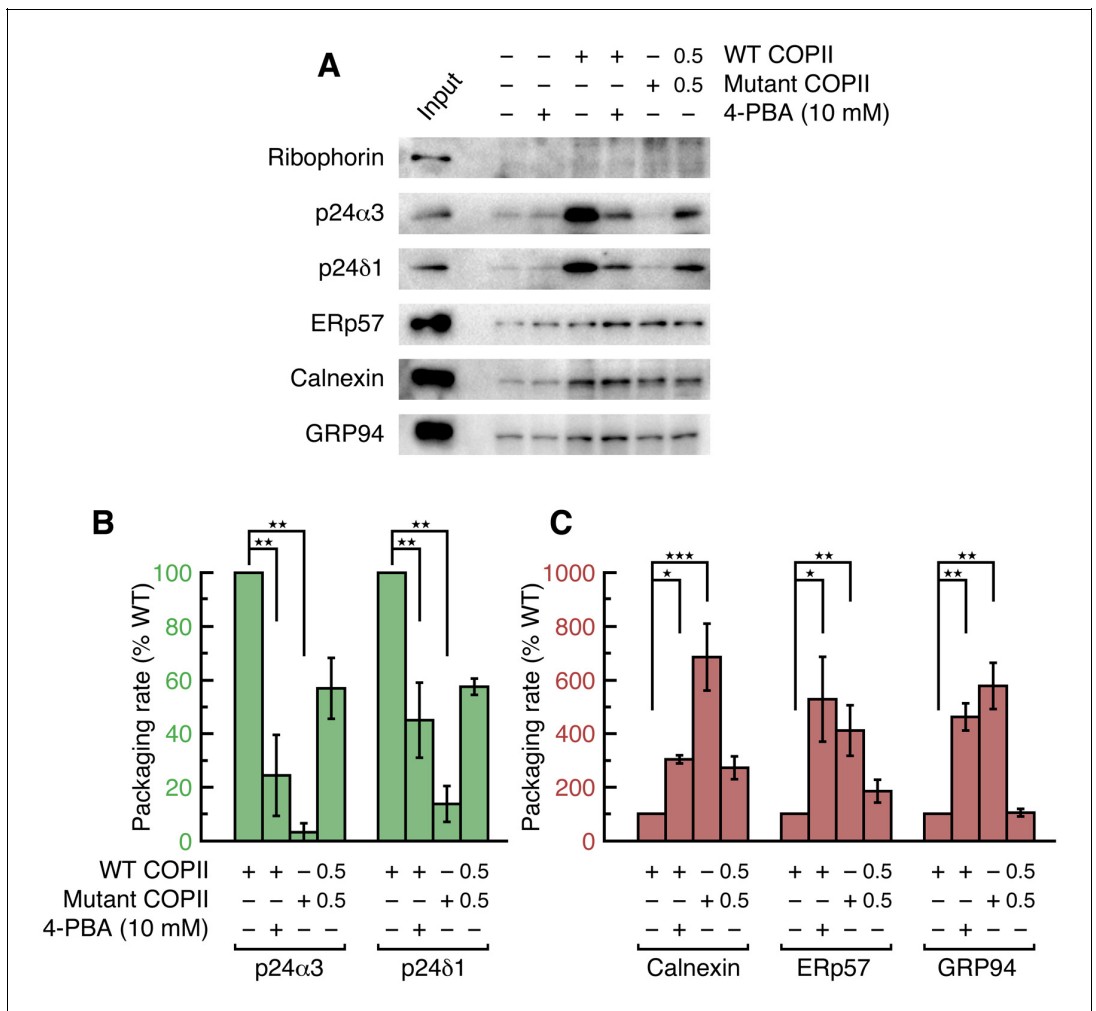

**Figure 4.** 4-PBA causes packaging of ER resident proteins into COPII vesicles budded from unstressed cell membranes. (**A**) Reconstituted COPII budding reactions were performed on permeabilized CHO cells (expressing neither wild-type nor mutant LDL receptor). Lane labeled Input represents 1.1% of the starting membranes used in a budding reaction; all other lanes represent the vesicle product of 100% of the starting membranes. Experiments were performed in triplicate. (**B**) Data in (**A**) were corrected as in *Figure 2D*, using the mean values of the packaging rates for Syntaxin 5, Erv46 and Rer1 (n = 3; ANOVA test, Bonferroni-Holm post hoc, **p<0.01). Bar graphs show mean ± s.e.m. (**C**) Graph shows packaging rates for Calnexin, ERp57 and GRP94 (n = 3; ANOVA test, Bonferroni-Holm post hoc, *p<0.05, **p<0.01 and ***p<0.001). Bar graphs show mean ± s.e.m.

levels (*Figure 4A* and *Table 3*). This latter observation is significant: it accords with the original finding of stringent exclusion of BiP from yeast COPII vesicles (*Barlowe et al., 1994*), and indicates that ER retention is properly imposed in our permeabilized-cell assay; and it highlights the contrasting finding that resident proteins are packaged at an elevated rate into wild-type COPII vesicles budded from stressed cells (*Figure 3C*). Quantification of these data in *Table 3* shows that wild-type COPII vesicles package resident proteins at a 5-fold higher rate from stressed cells than from unstressed cells, whereas the packaging rates of bona fide cargo proteins are essentially the same. In addition to the increased *rate* of packaging, it is also the case that UPR activation increases resident chaperone concentrations (though by how much is difficult to estimate since ER volume is expanded concomitantly (*Schuck et al., 2009*). The combination of increased levels (see, for example, *Figure 5*) and increased packaging rates of resident proteins that we observe in stressed cells yields substantial leakage into vesicles that is consistent with the observations of post-ER trafficking of residents and misfolded proteins cited above. The nature of the UPR-dependent change that is responsible for the increased leakage rate is unclear and will require further study. We note in passing that a sorting mechanism involving p24-dependent exclusion of residents should not be saturable, it should have a constant error rate of resident leakage, hence alternative processes should be considered such as UPR-induced changes in the size or polydispersity of resident-protein complexes.

In summary, from the analysis of budding experiments we infer two general effects concerning resident protein retention in the ER: a constitutive leakage in the UPR-activated cell; and the p24-dependent sorting of resident proteins from cargo to impose ER retention as a general property of COPII vesicle budding in stressed and unstressed cells.

## Extracellular secretion of the KDEL-tagged resident chaperone GRP94

The striking effect of 4-PBA on cell culture models of protein mistrafficking and ER stress has offered little clue as to the mechanism of action of the molecule (*Balch et al., 2008*). Although it is often referred to as a chemical chaperone, 4-PBA is active at concentrations far too low for it to be operating as an osmolyte. The alternative suggestion that 4-PBA mediates effects on trafficking of misfolded proteins as a histone deacetylase inhibitor (HDACi), made on the grounds of chemical similarity to the weak HDACi butyrate (*Rubenstein et al., 1997*), was refuted in a subsequent analysis (*Tveten et al., 2007*). The results we have presented thus far identify the COPII coat as a target of 4-PBA and suggest that its mechanism of action is to reduce the stringency of ER retention by inhibiting the packaging of p24 proteins into COPII vesicles. A prediction that follows is that 4-PBA application to whole cells should cause secretion of luminal resident proteins; this would occur if ER leakage of residents exceeds the capacity of Golgi-to-ER retrieval systems such as that involving the

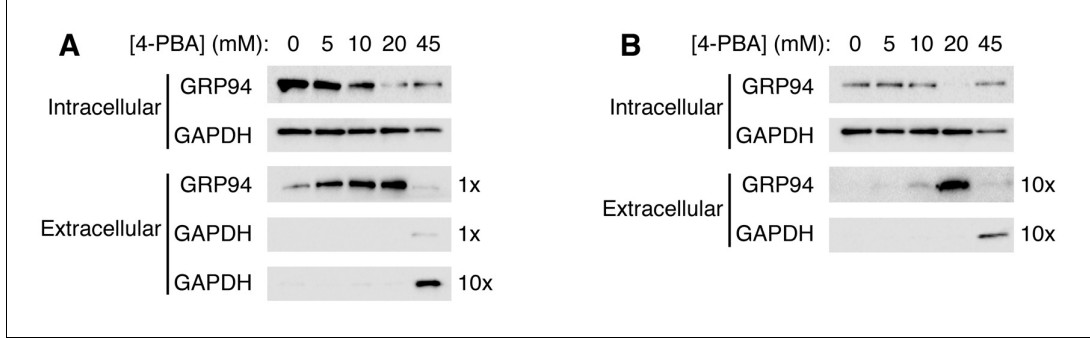

**Figure 5.** 4-PBA causes the secretion of the KDEL-tagged luminal resident GRP94. (**A**) CHO cells expressing G544V-mutant LDL receptor were incubated with increasing concentrations of 4-PBA for 24 hr at 37°C. Extracellular GRP94 in the medium was analyzed directly. Note that ER trapping of the mutant LDL receptor causes UPR leading to elevated levels of intracellular GRP94 (0 mM 4-PBA lane), and that UPR is attenuated at 10–20 mM 4-PBA (*Sørensen et al., 2006*). (**B**) Experiment used unstressed wild-type CHO cells. The blots in (**A**) and (**B**) showing intracellular levels of GRP94 are from equivalent amounts of cells (based on total protein measurement). To detect extracellular GRP94, ten times more sample was loaded than in (**A**).

KDEL receptor (*Dean and Pelham, 1990*). (Note that the KDEL receptor traffics independently of p24 proteins between ER and Golgi membranes [*Belden and Barlowe, 2001*]).

*Figure 5* presents the results of such an experiment. Cells were cultured with increasing concentrations of 4-PBA and then probed for the presence inside and outside cells of the KDEL-tagged luminal chaperone GRP94. We chose to analyze GRP94 because it is released into the medium by 4-PBA with little tendency to adhere to cell surfaces (unlike BiP and calreticulin), so could be measured directly without further manipulations or concentration. Strikingly, we observed secretion of GRP94, dose-dependent on 4-PBA and maximal at 20 mM 4-PBA, from both stressed and unstressed cells (*Figure 5A and B*, respectively). GRP94 is released by secretion rather than by cell lysis, because the cytosolic marker GAPDH is absent from the extracellular sample. At the highest concentration, 45 mM 4-PBA, cell lysis is detected and the cells have lost all ability to secrete GRP94. Thus, the profile of GRP94 secretion from stressed CHO cells closely matches that of the restoration of trafficking of the mutant LDL receptor (compare *Figures 3B* and *5A*). Importantly, the 4-PBA-dependent secretion of GRP94 is also observed from unstressed cells, even though significantly lower levels of GRP94 are present intracellularly at the outset of the assay (compare top rows in *Figure 5A and B*, and note that intracellular GRP94 levels in stressed cells are reduced over time as incubation with 4-PBA attenuates the UPR [*Sørensen et al., 2006*]). Thus, increasing concentration of 4-PBA—a treatment that lowers rather than raises UPR signaling levels—causes the secretion of a KDEL-tagged resident. Taken together with previous results, this supports the proposition that 4-PBA targets COPII to reduce the fidelity of ER retention.

The dose-response profiles of GRP94 secretion from stressed and unstressed cells are different (*Figure 5*). First, stressed cells secrete GRP94 in the absence of 4-PBA (albeit at low levels), matching a prior observation of BiP secretion from yeast cells following UPR induction with $\beta$-mercaptoethanol (*Belden and Barlowe, 2001*). Second, there is an apparent linear relationship between 4-PBA concentration and GRP94 secretion for stressed cells (*Figure 5A*). Since the KDEL retrieval system is known to be readily saturable (*Dean and Pelham, 1990*), these two observations indicate that resident protein leakage into COPII vesicles in stressed cells, even in the absence of 4-PBA, delivers KDEL ligands to post-ER membranes at levels that approach or exceed the capacity of the retrieval system. Third, the profile of GRP94 secretion from unstressed cells is a step response; only at high concentrations of 4-PBA does resident leakage seem to exceed a threshold level to cause saturation of the KDEL retrieval system (*Figure 5B*). Although the data are semi-quantitative, they suggest that protein sorting by COPII maintains resident leakage rates substantially below the threshold level, to render a useful dynamic range to the KDEL retrieval system under normal conditions. Implicit in this interpretation is that forward transport of the KDEL receptor substantially exceeds that of KDEL ligands, through the combination of receptor concentration (*Martínez-Menárguez et al., 1999*) and ligand depletion (*Figure 4*) in COPII vesicles. Generally, we interpret the behavior to indicate that ER residency is stringently imposed in unstressed cells by synergy of the COPII-dependent retention and KDEL-dependent retrieval mechanisms.

In summary, 4-PBA causes the extracellular secretion of a KDEL-tagged luminal resident in addition to restoring the trafficking of misfolded mutant LDL receptor, thereby phenocopying p24 mutant yeast and worms (*Elrod-Erickson and Kaiser, 1996*; *Wen and Greenwald, 1999*). The results indicate how ER residency may rely on the superposition of the retention and retrieval processes. Although our experiments focus on KDEL-mediated retrieval, it seems likely that additional retrieval systems, involving Rer1 and Erv41-Erv46, will likewise be supported by COPII-dependent retention (*Sato et al., 1995*; *Shibuya et al., 2015*). Finally, the saturation of the KDEL system that is indicated by our results in *Figure 5A* has relevance to the mechanism by which 4-PBA restores trafficking of misfolded LDL receptor. If escaped receptor molecules can be bound by chaperones such as BiP and retrieved to the ER via the KDEL receptor, as has been demonstrated for unassembled TCRα (*Yamamoto et al., 2001*), then restoration of mutant LDL receptor trafficking by 4-PBA may involve the combination of enhanced ER exit (*Figure 3*) and reduced retrieval upon saturation of the KDEL system.

## Sec24 protein as a target of 4-PBA

As a further test of COPII protein as a cellular target of 4-PBA, we analyzed a series of 4-PBA analogs (*Figure 6—figure supplement 1*) and compared the potency with which they restored trafficking of the mutant LDL receptor in the cell-based assay (*Figure 6A–D*) with their binding affinity for

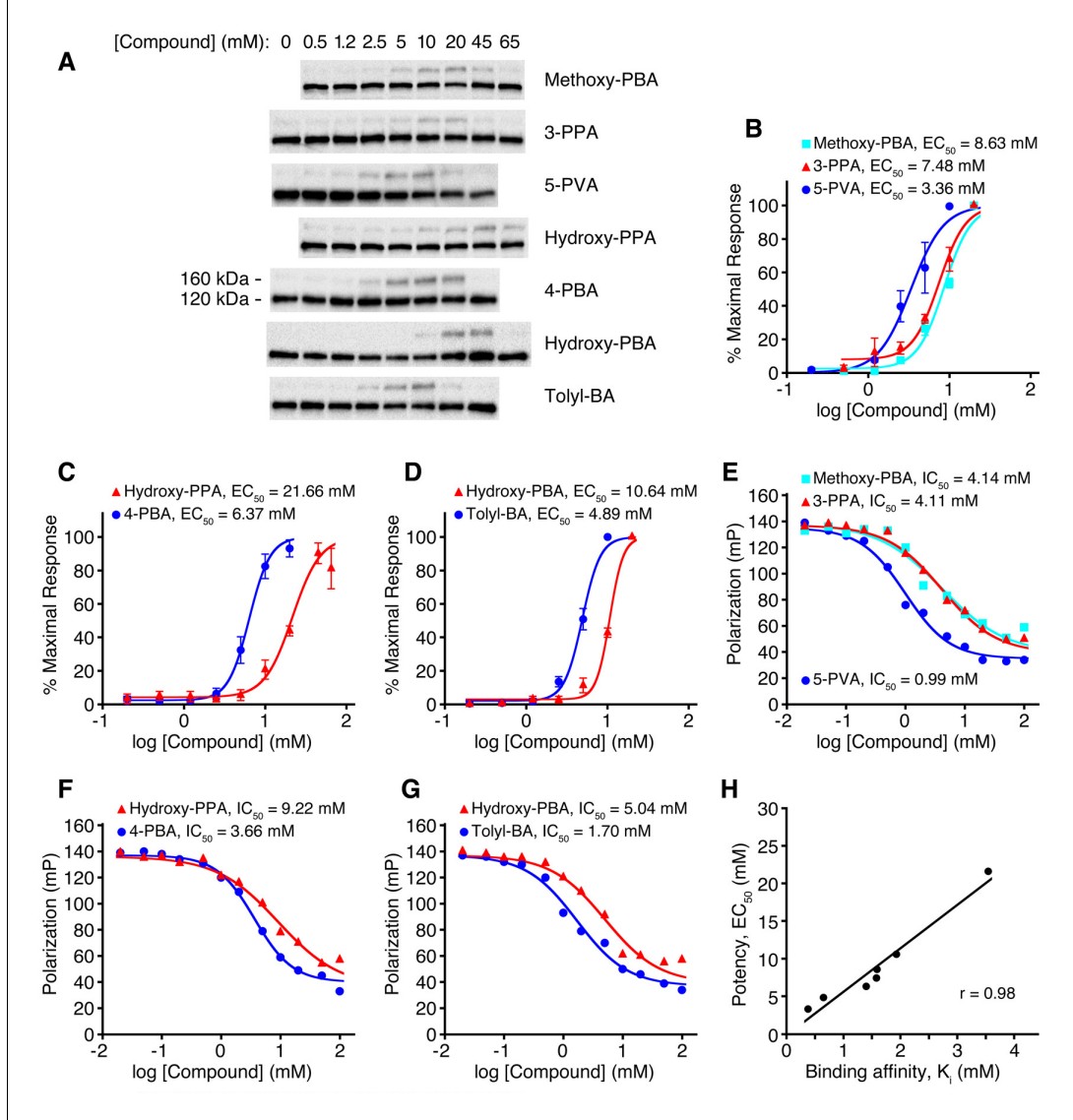

**Figure 6.** Relation between COPII binding-site affinity and pharmacological potency for a series of 4-PBA analogs. (**A**) CHO cells expressing G544V-mutant LDL receptor were incubated with the indicated concentrations of 4-PBA analogs for 2 hr at 37°C. Cell lysates were analyzed by immunoblotting for the appearance of 160 kDa mature glycosylated LDL receptor (labeling indicates the position of 160 kDa mature glycosylated and 120 kDa precursor forms of LDL receptor). A gamma correction has been applied to the picture to improve clarity. (**B**) The acquisition of mutant LDL receptor glycosylation as a function of compound concentration is plotted and $EC_{50}$ values reported. Data were obtained from (**A**) by densitometry of chemiluminescent signals. Methoxy-PBA is 4-(4-methoxyphenyl)butyrate (n = 4 independent experiments); 3-PPA, 3-phenylpropionate (n = 4); 5-PVA, 5-phenylvalerate (n = 5). Standard errors for $EC_{50}$ values are reported in Materials and methods. (**C**) Hydroxy-PPA, 3-(4-hydroxyphenyl)propionate (n = 4); 4-PBA (n = 5). (**D**) Hydroxy-PBA, 4-(4-hydroxyphenyl)butyrate (n = 4); Tolyl-BA, 4-(4-tolyl)butyrate (n = 4). (**E–G**) Displacement of the fluorescent probe 5-Fam-QIYTDIEANR (based on the ER export signal of VSV G protein) from human Sec24a by 4-PBA analogs was measured by fluorescence polarization at 22°C. $IC_{50}$ values were determined by non-linear regression fitting (standard errors are reported in Materials and methods). (**H**) Scatter plot compares data from the cell-based and fluorescence polarization experiments. For the latter, $IC_{50}$ values were converted to true affinities, $K_i$, using the approach of *Nikolovska-Coleska et al., 2004*. See also *Figure 6—figure supplement 1*.

The following figure supplement is available for figure 6:

**Figure supplement 1.** The series of 4-PBA analogs.

Sec24 protein (*Figure 6E–G*). For the latter, we used a competition binding assay based on fluorescence polarization that measures the affinity of 4-PBA analogs according to their ability to displace a fluorescent tracer peptide from Sec24a. The tracer is a modified DxE export sequence of VSV G protein, which binds to the B site (*Mancias and Goldberg, 2007*), hence the assay signal reports on specific binding to this site. The results of the two assays are compared in *Figure 6H*. Strikingly, the potency with which 4-PBA analogs restore trafficking of mutant LDL receptor increases in rank order of affinity for COPII protein. These results, together with the crystallographic observations and the vesicle budding experiments, provide multiple independent lines of evidence that 4-PBA targets the COPII coat.

## Conclusion

Using a biochemical strategy, with 4-PBA as a probe, we have identified and manipulated elements of the COPII coat that control protein sorting. The results of budding experiments show that COPII recognition of the ΦC motif mediates ~6 fold concentrative sorting of p24 proteins into vesicles, in line with previous measurements of 5–20-fold signal-mediated concentration of cargo and machinery molecules into COPII vesicles (*Balch et al., 1994*; *Malkus et al., 2002*; *Martínez-Menárguez et al., 1999*). Our data demonstrate that COPII can in addition exclude resident and misfolded proteins by a factor of 4–9-fold. Comparison with data on BiP suggests that this may be a reasonable measure of the magnitude of the ER retention process: a modified form of BiP lacking the KDEL retrieval signal is secreted from animal cells 5-fold more slowly than a protein tracer of bulk flow (*Barlowe and Helenius, 2016*; *Munro and Pelham, 1987*). Thus, by an active sorting mechanism COPII may concentrate cargo and deplete residents, to yield cargo purification of between 25- and 100-fold in a single budding event.

The subsequent purification step seems to occur at the intermediate compartment as anterograde cargoes are excluded from COPI budding vesicles that mediate retrograde traffic to the ER (*Martínez-Menárguez et al., 1999*). The underlying exclusion mechanism for COPI is unknown.

Stringent ER retention requires the efficient packaging of p24 proteins via the B site of the COPII coat. When signal-mediated packaging is prevented by B-site mutation, there remains an appreciable (25% of wild-type level) passive uptake into vesicles. However, we suggest that only those p24 proteins bound to the polymerized COPII coat can be organized into a luminal array capable of excluding resident protein complexes. The concept of a p24 array that drives resident-protein exclusion implies that coat polymerization and resident exclusion are coupled processes. Indeed, studies in yeast suggest that p24 deletion causes uncoupling. First, the principal p24 proteins Emp24p and Erv25p were identified as suppressors of the essential *sec13* gene, encoding a component of the COPII outer coat (*Elrod-Erickson and Kaiser, 1996*). Subsequent analysis indicated that Sec13 lends rigidity to the coat to assist the vesicle budding process (*Copic et al., 2012*). Thus, the finding that an impaired (△Sec13) COPII coat can bud vesicles from p24-depleted membranes, but not from wild-type membranes, yielded the surprising insight that p24 proteins mediate against vesicle formation in some manner; that is, the coat must do extra work in a wild-type p24 background (*Copic et al., 2012*; *Elrod-Erickson and Kaiser, 1996*). It was proposed that p24 proteins, by virtue of their asymmetric topology (i.e. predominantly luminal mass), increase the local membrane-bending energy and oppose curvature induction by the COPII coat (*Copic et al., 2012*). The results we have presented here suggest the additional or alternative explanation that the extra work is done as COPII self-assembly drives together the bound p24 proteins, and corresponds to force exerted thereby to displace resident complexes from the COPII bud. According to this speculative mechanical model, a macromolecule confined by an inward-moving p24 array would impart an outward-directed force on the p24 elements of the array that will depend strongly on macromolecule size, with the consequence that coat self-assembly can reject resident complexes to drive protein sorting. It will be of great interest now to determine the luminal form of p24 proteins in the COPII bud, and their distribution at ER exit sites generally. They are known to be the most abundant transmembrane constituents of COPII vesicles (*Otte et al., 2001*), but their stoichiometric relation to COPII protein, their oligomeric structure, and thus the nature of the putative array are all uncertain. A greater challenge is to understand the organization of ER chaperones. These seem to form highly dynamic, partially calcium-dependent complexes that are only marginally stable even at the very high macromolecule concentrations that prevail in the ER lumen (*Meunier et al., 2002*; *Tatu and Helenius, 1997*).

Finally, 4-PBA is FDA-approved as a prodrug for the treatment of urea cycle disorders; it is rapidly metabolized by $\beta$-oxidation to phenylacetate, which is conjugated with glutamine in the liver to provide an alternate route for waste nitrogen excretion (*Brusilow, 1991*). A growing body of evidence demonstrates the distinct property of 4-PBA to restore trafficking of misfolded proteins and to mitigate ER stress in a range of in vitro systems, and in animal models of metabolic and neurodegenerative diseases (*Bondulich et al., 2016*; *Ozcan et al., 2006*; *Rubenstein et al., 1997*). By establishing that 4-PBA acts on the COPII sorting machinery to attenuate ER retention, our findings should be an invaluable aid to proof of principle studies involving 4-PBA and its analogs.

## Materials and methods

### Expression vectors

The baculoviral constructs for overexpression of human COPII proteins in insect cells have been described previously (*Mancias and Goldberg, 2007*, *2008*). In the present study, we used constructs for H. sapiens Sec24a ($\triangle$1–340), full-length Sec24d and full-length Sec31a that were cloned into the vector pFast-HTB (Invitrogen, Carlsbad, CA), encoding a tobacco etch virus (TEV) cleavable N-terminal His$_6$ tag. Vectors encoding H. sapiens full-length Sec23a and Sec13a were of the type pFastBac1 (Invitrogen). The full-length Sec24d L750W mutant was generated using the Phusion site-directed mutagenesis kit (New England Biolabs, Ipswich, MA). One pair of 5'-phosphorylated primers (forward, TGGTA CACGA CAATC AGTGG TCAA; reverse, CACAG CACAC TGGAT TAAGG CTCC) was used to generate Sec24d L750W DNA from the wild type Sec24d template. Recombinant bacmids were generated in DH10Bac cells transformed with the various pFastBac vectors, using the Bac-to-Bac baculovirus expression system (Invitrogen).

DNA for full-length H. sapiens Sar1a was cloned into the vector pGEX-6P-1 (GE Life Sciences, Pittsburgh, PA), resulting in a protein with a N-terminal prescission protease (Invitrogen) cleavable glutathione S-transferase (GST) tag (protease cleavage yields the N-terminal sequence GPLGSMSFI...). DNA encoding H. sapiens Sec22b (residues 1–195) was cloned in the PET-28b vector (Novagen) with an additional N-terminal His$_6$-Smt3 fusion, as described (*Mancias and Goldberg, 2007*).

DNA encoding full-length H. sapiens LDL receptor (wild type and G544V forms) was cloned into the tetracycline-inducible expression vector pcDNA4/TO (Invitrogen) as described (*Sørensen et al., 2006*; *Tveten et al., 2007*). The G544V mutant LDL receptor was generated with Phusion site-directed mutagenesis kit, using two 5'-phosphorylated primers as follows: forward, GTCATC ACCCT AGATC TCCTC AGT; reverse, ATTGG GCCAC TGAAT GTTTT CAGT. The correct construction of all recombinant plasmid was confirmed by DNA sequencing (GENEWIZ, Inc, South Plainfield, NJ).

### Protein and peptide production

The expression in High Five insect cells (Invitrogen) and purification of COPII protein complexes were carried out essentially as described (*Mancias and Goldberg, 2008*). This includes the complex of Sec23a/Sec24a($\triangle$1–340) used for X-ray crystallography, Sec23a/Sec24d (full-length protein) for COPII budding assays, and Sec24a($\triangle$1–340) for fluorescence polarization experiments.

In preparation for protein production, recombinant baculoviruses were generated by transfecting Spodoptera frugiperda (Sf9) cells with bacmid DNA using the Cellfectin II Reagent (Invitrogen). Baculoviruses were amplified for three generations then used to infect High Five insect cells. Cells were harvested 48 hr post-infection and resuspended in a lysis buffer (50 mM Tris-HCl pH 8.0, 500 mM NaCl, 1 mM $\beta$-mercaptoethanol, 1 mM phenylmethylsulfonyl fluoride (PMSF) and protease inhibitor cocktail (Roche)).

Sec23a/Sec24a ($\triangle$1–340) and Sec24a ($\triangle$1–340) proteins were purified from cell lysates by HisTrap HP chromatography (GE Life Sciences) and the His$_6$ tag was removed using TEV protease (Invitrogen) at 4°C during overnight dialysis against a solution comprising 50 mM Tris-HCl pH 8.0, 300 mM NaCl and 5 mM Dithiothreitol (DTT). COPII proteins were diluted 3-fold with a low-salt buffer (50 mM Tris-HCl pH 8.5, 50 mM NaCl and 5 mM DTT) and loaded into a 5 ml HiTrap Q column (GE Life Sciences) followed by gradient elution with 50 mM Tris-HCl pH 8.0, 250 mM NaCl and 5 mM DTT. Finally, the protein was purified by size exclusion chromatography on a HiLoad 26/60 Superdex 200 column (GE Life Sciences) in 20 mM Tris-HCl pH 7.5, 200 mM NaCl and 4 mM DTT.

The full-length Sec23a/Sec24d proteins, wild type and L750W mutant, and Sec13a/31a protein used in budding experiments were purified in the same manner, except that N-terminal His$_6$ tags were not removed. Following the affinity purification step, proteins were diluted directly (rather than dialyzed) into low salt buffer, followed by chromatographic purification using HiTrap Q followed by HiLoad 26/60 Superdex 200 columns in buffer 20 mM Tris-HCl pH 7.5, 500 mM NaCl and 1 mM DTT.

The full-length H. sapiens Sar1a protein was expressed in E.coli BL21(DE3) (Invitrogen) in Luria-Bertani (LB) broth. At a culture density of OD$_{600}$ = 0.8, protein expression was induced by the addition of 0.2 mM isopropyl $\beta$-D-1-thiogalactopyranoside at 16°C overnight. GST-tagged Sar1a was purified first on a GST-Trap 4B column (GE Life Sciences) in PBS buffer (10 mM Na$_2$HPO$_4$ pH 7.4, 137 mM NaCl, 2.7 mM KCl, 1 mM PMSF). The GST tag was removed with prescission protease and Sar1a was purified further by size exclusion Superdex 75 chromatography in a buffer containing 20 mM Tris-HCl pH 7.5, 500 mM NaCl and 1 mM DTT, as described (*Kim et al., 2005*).

The cytosolic region (residues 1–195) of H. sapiens Sec22b was purified as described (*Mancias and Goldberg, 2007*). Briefly, the protein was expressed in E.coli with an N-terminal His$_6$-Smt3 tag. Following removal of the tag with Ulp1 protease, protein was purified on Hi-Trap Q column, then by size exclusion Superdex 75 chromatography in 20 mM Tris-HCl pH 7.5, 200 mM NaCl and 4 mM DTT. All purified proteins were concentrated, aliquoted, flash frozen and stored at −80°C.

Synthetic peptides containing the ΦC signal motif, at >95% purity, were purchased from the Tufts University Core Facility (see *Table 1* for full peptide sequences).

## Protein crystallization and structure determination

The Sec23a/Sec24a•Sec22b complex was crystallized as described previously (*Mancias and Goldberg, 2007*) from a solution comprising 10% (w/v) PEG 4000, 500 mM Na acetate and 50 mM Tris-HCl pH 7.9. Since acetate ions compete with ligand binding at the B site, crystals were transferred into a soaking buffer of 10% (w/v) PEG 4000, 600 mM NaCl, 50 mM Tris-HCl buffer, pH 7.9. (Two acetate ions are observed bound to B-site arginine residues in electron density maps calculated from Sec23a/Sec24a•Sec22b crystallographic data, PDB 2NUT). To this was added 4-PBA (0–50 mM) or 5 mM ΦC signal peptide, and ligand soaking proceeded for five hours at 22°C. Crystals were then transferred to an equivalent ligand-containing buffer with additional 24% ethylene glycol cryo-protectant, and were flash frozen in liquid nitrogen. Crystals treated in this manner diffracted synchrotron X-rays to 2.5–2.8 Å resolution.

X-ray diffraction data were collected at 0.979 Å wavelength on beamline ID-24C or ID-24E of the Advanced Photon Source, Argonne National Laboratory. Data were processed with the program HKL2000 (*Otwinowski and Minor, 1997*; *Tables 1* and *2*). Structure refinement was carried out using Phenix (*Adams et al., 2010*), initially by rigid-body improvement, followed by positional refinement, based on the published coordinates of Sec23a/Sec24a•Sec22b (PDB 2NUT; *Mancias and Goldberg, 2007*). Models for ΦC signal peptides or 4-PBA were not included in refinements prior to the creation of *Figure 1*, to eliminate potential phase bias. These ligands were modeled in a final stage of refinement, the statistics for which are summarized in *Tables 1* and *2*.

## Cell lines and cell-based experiments

T-Rex-CHO-K1 cloned cell lines expressing wild-type and mutant LDL receptor were cultured as described (*Sørensen et al., 2006*; *Tveten et al., 2007*), and according to the manufacturer's protocols (Thermo Fisher Scientific; CHO cell Lot No. 1610593). The expression in CHO cells of G544V-mutant or wild-type LDL receptor protein was induced with 1 μg/mL tetracycline for 24 hr prior to experiments with 4-PBA or 4-PBA analogs (*Tveten et al., 2007*). To compare pharmacological potency of 4-PBA analogs, cells were incubated with the compounds for 2 hr at 37°C in 12-well 22 mm plates (Corning Costar #3513). For experiments concerning the extracellular secretion of GRP94 (*Figure 5*), the incubations with 4-PBA were extended to 24 hr. Cells were broken in lysis buffer (1% (v/v) Triton X-100, 150 mM NaCl, 10 mM EDTA, 50 mM Tris-HCl pH 7.5) and cleared by centrifugation (15,000 g for 15 min). Cellular protein concentration was measured using 'Protein Assay Kit II' (Bio-Rad) with bovine serum albumin as standard. Lysates were added to Laemmli sample buffer (New England BioLabs) containing 130 mM dithiothreitol and heated for 15 min at 55°C. Cell

medium (containing secreted proteins) was added to Laemmli sample buffer containing 5% $\beta$-mercaptoethanol and boiled for 5 min. Samples were run on 4–15% Tris/HCl Criterion gels (Bio-Rad) followed by immunoblotting.

## Permeabilization of CHO cells

In vitro COPII budding reactions used permeabilized cells prepared from the T-Rex-CHO-K1 parent line and from the clone that expresses mutant LDL receptor. Permeabilization was carried out as described (*Mancias and Goldberg, 2008*) with modifications. Two culture dishes (150 mm) of cells were prepared with cells at no greater than 60% confluence. Cells were washed with 20 ml PBS at 37°C and then trypsinized with 2 ml of Trypsin-EDTA (0.05%, Gibco) for 1 min at 37°C; 100 µl of Soybean Trypsin Inhibitor (2 mg/ml; Sigma) was added to each culture plate. To this was added 5.5 ml ice-cold KHM buffer (110 mM KOAc, 20 mM Hepes pH 7.2, 2 mM magnesium acetate) per plate, and cells were transferred to a 15 ml Falcon tube and pelleted gently (250 g for 3 min at 4°C). The supernatant was aspirated and the cells resuspended in 6 ml ice-cold KHM buffer. To permeabilize the cells, 6 µl digitonin stock solution (40 mg/ml in dimethyl sulfoxide) was added, cells were mixed by inversion and incubated on ice for 5 min. To terminate permeabilization, cells were diluted to 14 ml with KHM buffer and immediately pelleted at 250 g for 3 min at 4°C. Cells were resuspended in 14 ml ice-cold HEPES buffer (50 mM HEPES pH 7.2, 90 mM potassium acetate), incubated for 10 min on ice and harvested at 250 g for 3 min at 4°C. Finally, the cells were resuspended in 1 ml of KCLM buffer (110 mM potassium Chloride, 20 mM HEPES pH 7.2, 2 mM magnesium chloride), harvested at 10,000 g for 15 s and resuspended in 60 µl of KCLM buffer. The protein concentration was measured using protein assay kit (Bio-Rad) with BSA as standard; routinely, the measurement is in the range 4–5 mg/ml. Permeabilized cells were used in budding experiments immediately following their preparation.

## In Vitro COPII budding assay

The COPII vesicle budding assay was carried out as described (*Mancias and Goldberg, 2008*) with minor modifications. Each budding reaction mixture (100 µl) contained 110 mM potassium chloride (chloride was used in place of acetate ions which interfere at the B site), 20 mM HEPES pH 7.2, 2 mM magnesium chloride, protease inhibitor cocktail (1X Roche cOmplete), 0.2 mM GTP, ATP regeneration system (40 mM creatine phosphate, 0.2 mg/ml creatine phosphokinase, 1 mM ATP) and, where indicated, 10 mM 4-PBA. This mixture was brought to 37°C, permeabilized cells (25 µg) were added and the mixture incubated for 1 min at 37°C. The budding reaction was initiated by the addition of coat proteins (2 µg of Sar1a, 2 µg Sec23a/24d or Sec23a/24d-L750W, 4 µg Sec13a/31a) and budding proceeded for 10 min at 37°C. A short budding reaction time was employed—previously 30–40 min (*Mancias and Goldberg, 2007*)—to avoid potential contamination of vesicle product with post-ER membranes. Also, recombinant Sec24d was used for all budding experiments since a sufficiency of Sec23a/24d and Sec23a/24d-L750W proteins, but not of other paralogs, could be prepared for scaled-up budding reactions. Reactions were terminated at 4°C and centrifuged at medium speed (12,000 g) for 20 min at 4°C. 90 µl of supernatant was transferred to polypropylene centrifuge tubes (Beckman, #343621) and vesicles were pelleted at 55,000 rpm for 35 min at 4°C in a Beckman TLA 100.1 rotor. To the pellet, 7 µl of lysis buffer (10 mM Tris-HCl pH 7.5, 100 mM NaCl, 1% (v/v) Triton X-100, 300 mM DTT) and 3.5 µl of 3X Laemmli sample buffer were added. The samples were heated at 42°C for 10 min and 10 µl sample was loaded on 4–20% Tris/HCl Criterion gels (Bio-Rad). Subsequent analysis was carried out by immunoblotting.

## Fluorescence polarization assay

The fluorescence assay was based on the approach described previously using yeast COPII proteins (*Mossessova et al., 2003*). A synthetic peptide (sequence QIYTDIEANR, based on the DxE export signal of VSV G protein, shown previously to bind specifically the B site [*Mancias and Goldberg, 2008*]) conjugated at the N terminus with 5-carboxyfluorescein (5-Fam) was purchased from the Tufts University Core Facility (>95% purity); a 250 µM stock solution in Fluor buffer (160 mM NaCl, 50 mM HEPES-NaOH pH 7.5) was stored at −80°C. Stock solutions of 4-PBA analogs (highest purity available) were 714 mM, adjusted to pH 7.5 with NaOH. Fluorescence polarization titrations were performed on a FluoroMax-4 spectrofluorometer with autopolarizer (Horiba Scientific). Excitation and

emission wavelengths were set to 490 and 520 nm, respectively, and slits were adjusted to 2 nm to yield approximately $1.5 \times 10^6$ counts during a 0.1 s signal acquisition with the sample held in a 45 μl cuvette (Hellma analytics cuvette 105.251). The affinity between 5-Fam-QIYTDIEANR and Sec24a was measured by titration as $K_d$ = 14.2 μM. On this basis, competition binding titrations were carried out by maintaining constant concentrations of 5-Fam-QIYTDIEANR and purified Sec24a (5 μM and 22 μM, respectively), while the concentration of the competitor 4-PBA analog was increased in the range 0.2–100 mM to displace the fluorescent peptide. This ratio of fluorescence reporter concentration, Sec24a concentration, and 14.2 μM dissociation constant ensures that the majority of the reporter will be bound to Sec24a at the beginning of the titration. The relatively low concentration of Sec24a (~1.5 times the $K_d$ for the reporter) has the effect of reducing the dynamic range of the assay. However, high concentrations of Sec24a would increase the $IC_{50}$ for all 4-PBA analogs to excessively high values. Under the conditions chosen, the dynamic range is ~100 mP (*Figure 6*).

All titrations were carried out at 22°C in degassed Fluor buffer. Reaction mixtures were incubated at room temperature for 10 min then centrifuged at 15,000 g for 10 min prior to fluorescent measurement. Positive controls containing free fluorescent reporter without Sec24a (equivalent to 100% inhibition) were included for each titration.

Note that synthetic peptides containing the DxE motif of VSV G protein bind to Sec24a/b but not to the Sec24c/d paralogs. Hence, our experiments involving binding and peptide competition at the B site were restricted to fluorescence experiments with Sec24a/b. No high-affinity (< 100 μM) peptide ligands are available that bind to Sec24c/d, hence peptide-competition analysis was not possible in our budding experiments.

## Quantification and statistical analysis

### Immunoblot quantification

Proteins were electrotransferred from SDS-PAGE gels onto Immun-Blot low fluorescence PVDF membrane (Bio-Rad) and blocked using TBST (5% BioRad Blocker in 25 mM Tris-HCl pH 7.4, 150 mM NaCl, 0.05% (v/v) Tween 20). Membranes were incubated at 4°C overnight with primary antibodies in TBST followed by incubation with horseradish peroxidase-conjugated secondary antibodies. Chemiluminescence signals were generated by incubation with ECL Prime Western Blotting detection reagent (GE Healthcare #RPN2236). The COPII budding experiments involving unstressed cells (*Figure 4*) yielded lower protein levels, hence we used the high-intensity ECL Select reagent for these blots (GE Healthcare #RPN2235) and the analysis was restricted to the highest quality antibodies. Chemiluminescence signals were detected using a ChemiDoc MP Imaging System and acquired with Image Lab 5.0 software (Bio-Rad). Quantification of protein band intensities was performed using Image Gauge software Version 4.1 (Fujifilm), including background correction of the raw image data.

For analysis of COPII budding, the background-subtracted packaging measurements (for p24 proteins, resident proteins and G544V-mutant LDL receptor) were corrected for change in budding yield due to treatment (i.e. 10 mM 4-PBA or mutant COPII protein) using the mean values of the packaging measurements for Syntaxin 5, membrin, Erv46 and Rer1. The datasets in triplicate were placed on a common scale (by minimizing the mean of the ratios of pairwise lane differences). For graphical presentation, the data were then normalized and presented as a percentage of the budding rate due to wild-type COPII coat. However, all statistical analysis was performed on data prior to normalization in order to preserve statistical distributions.

### Binding analysis for 4-PBA ligands

Chemiluminescence signals from western blots for the glycosylated (160 kDa) form of G544V-mutant LDL receptor were quantified as a percentage of total LDL receptor protein (glycosylated + unglycosylated bands). Data at high concentrations of 4-PBA analogs, which show inhibition of LDL receptor trafficking, were omitted. Data from individual experiments were normalized (to obtain maximal response = 100%), so that data from replicate experiments (n = 4–5) could be averaged. Logistic fitting to the dose-response data was carried out with program Ultrafit (Elsevier Biosoft). The $EC_{50}$ values and associated standard errors for the fits shown in *Figure 6B–D* are as follows: Methoxy-PBA, $8.63 \pm 0.98$ mM; 3-PPA, $7.48 \pm 0.68$ mM; 5-PVA, $3.36 \pm 0.42$ mM; Hydroxy-PPA, $21.66 \pm 1.03$ mM; 4-PBA, $6.37 \pm 0.50$ mM; Hydroxy-PBA, $10.64 \pm 0.40$ mM; Tolyl-BA, $4.89 \pm 0.24$ mM.

Data from fluorescence polarization titrations (*Figure 6E–G*) were analyzed with Ultrafit, and $IC_{50}$ values were determined from plots using nonlinear curve fitting. The $IC_{50}$ values and associated standard errors for the fits in *Figure 6E–G* are as follows: Methoxy-PBA, $4.14 \pm 0.70$ mM; 3-PPA, $4.11 \pm 0.34$ mM; 5-PVA, $0.99 \pm 0.08$ mM; Hydroxy-PPA, $9.22 \pm 0.94$ mM; 4-PBA, $3.66 \pm 0.22$ mM; Hydroxy-PBA, $5.04 \pm 0.69$ mM; Tolyl-BA, $1.7 \pm 0.16$ mM. $IC_{50}$ values were converted to true affinities, $K_i$, using the approach of *Nikolovska-Coleska et al., 2004*, and these are used in the scatter plot, *Figure 6H*.

## Statistics

The statistical parameters and numbers of replicate measurements are reported in figure legends. All instances of replicate measurements are biological replicates; that is, they are measurements of biologically distinct samples. For analysis of COPII budding experiments, chemiluminescent densitometry data were assembled in Excel and group means were compared by one-way ANOVA. A post-hoc Bonferroni-Holm test was used to control the experiment-wise error rate for a subset of the pairwise comparisons, specifically the cargo-packaging rate due to COPII relative to 4-PBA treatment or to mutant-COPII treatment. The 50/50 (wild-type/mutant) COPII mix was used in budding experiments merely to emulate the effect of 10 mM 4-PBA; this mix has a partial effect on cargo packaging, with reduced statistical power relative to 100% mutant COPII, hence these data were excluded from statistical analysis.

As reported in *Figure 6H*, the comparison between COPII binding-site affinity and pharmacological potency for the series of 4-PBA analogs yields a linear correlation coefficient $r(5) = 0.98$; this is significant at $p < 0.0005$ for a two-tailed test.

## Acknowledgements

We are grateful to Wim Annaert and Charlie Barlowe for gifts of antibodies to Rer1 and Erv46, respectively. Crystallographic experiments were carried out at NE-CAT beamlines at the Advanced Photon Source of the Argonne National Laboratory (these facilities were supported by NIH Grants ACB-12002, AGM-12006, P41 GM103403, and S10 RR029205, under Department of Energy Contract DE-AC02-06CH11357). This research was supported in part by the Memorial Sloan Kettering Cancer Center Core Grant P30-CA008748.

## Additional information

### Funding

| Funder | Grant reference number | Author |
| --- | --- | --- |
| Howard Hughes Medical Institute | | Jonathan Goldberg |
| Memorial Sloan-Kettering Cancer Center | P30-CA008748 | Jonathan Goldberg |

The funders had no role in study design, data collection and interpretation, or the decision to submit the work for publication.

### Author contributions

WM, EG, Data curation, Formal analysis, Validation, Investigation, Visualization, Methodology, Writing—review and editing; JG, Conceptualization, Formal analysis, Supervision, Funding acquisition, Investigation, Writing—original draft, Writing—review and editing

### Author ORCIDs

Jonathan Goldberg, http://orcid.org/0000-0001-9439-321X

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
