## [Decision Letter]

Thank you for submitting your article "ER retention is imposed by COPII protein sorting and attenuated by 4-phenylbutyrate" for consideration by *eLife*. Your paper has been seen by three reviewers, two of whom are experts in the field. The third reviewer, David Ron (Reviewer #1), is a non-expert member of our Board of Reviewing Editors and Vivek Malhotra serves as the Senior Editor. The following individuals involved in review of your submission have agreed to reveal their identity: Elizabeth A Miller (Reviewer #2); Charles Barlowe (Reviewer #3).

The reviewers have discussed the reviews with one another and the Reviewing Editor has drafted this decision to help you prepare a revised submission.

Summary:

All three reviewers recognize the significance of your discovery of 4-PBA binding site in B-pocket of Sec24 and of the mechanistic insight it provides into the ability of this compound to favour ER exit of proteins that would otherwise be retained in the organelle (ERQC substrates) – an activity previously attributed to 4-PBA's activity as a chemical chaperone.

Essential revisions:

1) The reviewers felt that the proposal that 4-PBA exerts its effects on ER export by competing for B-site ligands that are present on P24 class of cargo receptors is in need of further experimental validation. The reviewers consider that this could be achieved by correlating the affinity for the B-site of 4-PBA analogues (and/or peptide mimetics of B-site ligands on cargo receptors) with their ability to loosen ER quality control in your in vitro workhorse assay. A good correlation will support your model and a lack of correlation will speak against it.

Alternatively, Sec24 might be over-expressed to test the prediction that this reverses the effects of 4-PBA in vivo.

2) The method used to normalize the effect of 4-PBA on ER export – a method that led you to conclude that 4-PBA selectively promotes the export of ERQC substrates – came under scrutiny. Reviewer 2 notes that not everyone observes the gross defect in vesicular transport reported on here. In the course of the consultation, reviewer 3 suggested that increasing the concentration of Sec23/Sec24 in the cell free reaction might overcome this issue. Reviewer 1 wonders how such a gross defect is consistent with a net increase in export of ERQC clients observed in cells treated with 4-PBA (modelled by LDL-R here, in vitro)? Whilst we are unable to offer a specific experiment to address this issue, we feel it is a problem of sufficient magnitude that we are happy to shift the onus to resolve it on to you.

Reviewer #1:

This paper reports on the discovery that 4-phenyl butyrate engages a site on the SEC24 subunit of COPII coat normally reserved for cargo receptors and goes on to reveal a surprising equivalence between the defect thereby imposed on cargo incorporation into vesicles in an in vitro preparation exposed to 4-PBA and one constituted with COPII coat mutated in the cargo receptor site. Further evidence is provided to support the idea that 4-PBA's ability to encourage the trafficking of misfolded secreted protein (normally retained by ER quality control, ERQC, apparatus) derives from its ability to block cargo incorporation into export vesicles. These findings shed light both on fundamentals of the constitution of ERQC apparatus and on the action of a mysterious if chemically simple compound.

Specific comments:

Figure 6 correlates the affinity of 4-PBA analogues for the B site of SEC24 with their potency in encouraging trafficking of mutant LDL out of the ER in live cells. An extension of this correlation to the effects of an informative subset of these analogues in the in vitro budding assays would provide further confidence in the model (whereby neutering cargo receptor binding to COPII contributes to 4-PBA's effect on client protein trafficking).

In a similar vein (and perhaps this has already been noted previously): Given that 4-PBA is a low affinity ligand for the B site; has it been shown that higher affinity ligands also work against incorporation of cargo and favor incorporation of ER residents and misfolded protein into vesicles that bud out of the ER in the assays shown in Figure 2–Figure 4? If not, such an experiment should be added in validation.

Plotted in 4C is the relative recovery of Cnx/Grp94/ERp57 etc normalized to P24. This indeed showcases nicely the selective effect of 4-PBA and mutant Sec24 on packaging of cargo receptors, but given the overall decline in vesicle budding induced by the chemical, it is hard to understand how its application results in a net increase in flow of resident ER and (presumably misfolded) client proteins. Are there regulatory mechanisms operative in vivo that defend total vesicular transport out the ER despite the defect imposed by 4-PBA or the mutant SEC24/COPII?

Reviewer #2:

This manuscript from the Goldberg lab explores the mechanism of action of the putative chemical chaperone, 4-PBA, in mediating ER quality control events. Using crystallographic analysis, the authors find that both 4-PBA and the p24 family of proteins interact with the well-characterized B-site on the COPII cargo adaptor protein, Sec24. Occlusion of this site by mutation, or treatment with 4-PBA in vitro, leads to reduced packaging of p24 proteins and concomitant increase in packaging of ER residents and misfolded LDL-R. These data change the interpretation of the mechanism of action of 4-PBA and thus provide a broad and important insight into mechanisms of protein quality control within the secretory pathway. In general, the findings are interesting and the model presented is exciting. However, there are some concerns, although I don't necessarily have good suggestions for all of them!

1) The authors elide the somewhat cloudy issue of the precise export signal on p24 proteins, and cite only the Nakamura and Nufer papers that delineate the C-terminal ΦC motif. Belden and Barlowe, however, published a FF/YF motif in the yeast proteins that was shown to interact directly with COPII (both Sec23/24 and Sec13/31, somewhat confusingly). The utility of this signal was then demonstrated in vivo by our lab (D'Arcangelo et al. 2015). Our own findings that looked at packing and mutual protein stability suggest that the FF/YF motif is a bona fide export signal, but of course don't rule out that there is an additional contribution from the ΦC motif. It's tempting (for me) to speculate that the two signals have subtly different affinities for the different Sec24 isoforms that might be of relevance here and in the yeast work that suggests ligand-binding alters affinity for different Sec24 isoforms (Muniz lab). So, I appreciate that the authors don't fully believe the Barlowe/Miller lab findings and simply avoid this discussion rather than being dismissive. However, for a reader not intimately familiar with the research history, this omission makes the field more confusing.

2) In relation to the above point, I note that the peptides used for crystallography only encompass the C-terminal 6 residues, therefore lack the FF/YF motifs described above. In many species, these motifs are flanked by an acidic Glu residue: is it possible that this could be accommodated in the B-site and contribute to the charge interactions that the carboxylate mediates in the ΦC motif?

3) The authors have previously clearly demonstrated the mechanism of cargo discrimination by Sec24a/b and Sec24c/d isoforms at the B-site. Now they show that p24 proteins and 4-PBA are equal substrates for all isoforms. The mechanism by which this differs from the other peptide interactions at the B-site should be more fully explained/explored.

4) I'm very uncomfortable with the method of normalization of vesicle budding experiments. Here, I'm afraid I don't have a good alternative suggestion, given the realities of the experiments. I would note, however, that when we and others (Wieland lab) have used mutant Sec24 isoforms in budding experiments, we generally don't see a decrease in vesicle release. This has always been an important indicator that the mutations don't perturb COPII function more broadly but simply report on cargo enrichment. So, I don't fully understand why there is such a dramatic reduction in vesicle formation. I also don't think that normalization for vesicle release by monitoring an "unaffected" cargo protein is the best solution to this problem. Again, in the past, we and others have used lipid release as a measure of vesicle formation.

5) Is there an explanation for why ER resident packaging in the 50/50 wt/mutant COPII condition is not equivalent to the 4-PBA treatment? I thought that the point of this condition was to mimic that of the 4-PBA in terms of degree of reduction of p24 packaging. If p24 depletion is the same and vesicle release is equivalent, shouldn't capture of ER residents also go up? It might be a good idea to show the normalization blots for each experiment.

6) I have one suggestion for an experiment to validate the proposed model and substantiate the correlation between 4-PBA effects and Sec24 inhibition: the authors could overexperess Sec24 in cells, both wt and mutant, and repeat the 4-PBA titration and LDL-R/GRP94 leakage assays.

7) Finally, I really like the steric model proposed by the authors as to how p24 proteins might act to exclude ER resident proteins. I note that this concept was also proposed by Chris Kaiser (PNAS 2000) and the steric effects of cargo crowding noted in a review by Stachowiak/Brodsky/Miller (Nature Cell Biol. 2013).

Reviewer #3:

In this manuscript Goldberg and colleagues investigate the mechanism by which 4-phenylbutryate (4-PBA) improves trafficking of misfolded secretory proteins. There are two important advances reported here. First, a molecular target of 4-PBA was defined in the Sec24 B-site, a cargo-binding site that recognizes and selects specific sorting signals for uptake into COPII vesicles. The Sec24 B-site is necessary for efficient uptake of multiple cargo including the abundant p24 family of proteins. This leads to a second important advance wherein packaging of p24 proteins into COPII vesicles can be acutely inhibited in cell free budding reactions to then monitor influences on cargo sorting into COPII vesicles. Treatment with 4-PBA, or use of a Sec24 B-site mutant, inhibited p24 packaging whereas the levels of misfolded cargo (mutant LDL-R) and ER resident proteins (e.g. calnexin and ERp57) were increased in COPII budded vesicles. The authors conclude that the COPII machinery in concert with p24 proteins can actively exclude ER-resident proteins and misfolded cargo from forming vesicles.

The experimental results provide strong support for their conclusions. The discussion is thorough with appropriate caution on interpretation. I have no concerns with this excellent study.

---

## [Author Response]

*Essential revisions:*

*1) The reviewers felt that the proposal that 4-PBA exerts its effects on ER export by competing for B-site ligands that are present on P24 class of cargo receptors is in need of further experimental validation. The reviewers consider that this could be achieved by correlating the affinity for the B-site of 4-PBA analogues (and/or peptide mimetics of B-site ligands on cargo receptors) with their ability to loosen ER quality control in your* in vitro *workhorse assay. A good correlation will support your model and a lack of correlation will speak against it.*

*Alternatively, Sec24 might be over-expressed to test the prediction that this reverses the effects of 4-PBA* in vivo.

Regarding the use of 4-PBA analogs in the in vitro assay: Because 4-PBA is such a simple, low-complexity molecule, the homologous series of analogs available to us (Figure 6—figure supplement 1) involves closely related compounds that differ in affinity and potency by just one order of magnitude across the range. Our fluorescence polarization assay (Figure 6) is easily able to discern these differences, as is the cell-based assay when appropriate numbers (n=4–5) of replicate measurements are made (Figure 6). While the statistical power is clearly sufficient for the high-dose 4-PBA budding experiments (Figure 2–Figure 4), as confirmed by the post-hoc tests, the in vitro budding assay is inadequate for the dose-response experiment (suggested by the Reviewing Editor) using 4-PBA analogs—observe, for example, the standard errors for the effect of 4-PBA treatment on calnexin and ERp57 packaging in Figure 3.

In our view, Figure 6 reports on a superior experiment because it compares the physical chemistry of binding involving pure COPII proteins to the effect on trafficking in whole cells (rather than to in vitro budding which would repeat the use of purified COPII). The results show that the incremental changes in chemical shape and polarization in the homologous series alter binding affinity for Sec24a and shift the potency for mutant LDL receptor trafficking with striking correlation. In the context of a target validation analysis, this should be considered a gold-standard experiment. But we have done far better than this one test of the 4-PBA target and mechanism of action. Our study includes three additional tests, namely the crystallographic demonstration that 4-PBA and the ΦC signal bind to a common site on COPII (Figure 1 and Figure 1—figure supplement 1), the budding experiments revealing predicted effects of 4-PBA (Figure 2–Figure 4), and the GRP94 secretion experiment (Figure 5).

The experiment suggested by reviewer 2, involving Sec24 over-expression, is not appropriate here for two reasons. First, this is not a test for a mechanism-of-action target, since any protein that binds to ligand will cause ligand displacement from the cellular target. Second, 4-PBA binds with millimolar affinity, and comparable levels of Sec24 would be required in the cell; for example, to observe a 30–50% shift in 4-PBA potency in our assays, Sec24 would have to be over-expressed in cytosol at a concentration of 50–100 mg/ml, clearly unfeasible.

*2) The method used to normalize the effect of 4-PBA on ER export – a method that led you to conclude that 4-PBA selectively promotes the export of ERQC substrates – came under scrutiny. Reviewer 2 notes that not everyone observes the gross defect in vesicular transport reported on here. In the course of the consultation, reviewer 3 suggested that increasing the concentration of Sec23/Sec24 in the cell free reaction might overcome this issue. Reviewer 1 wonders how such a gross defect is consistent with a net increase in export of ERQC clients observed in cells treated with 4-PBA (modelled by LDL-R here,* in vitro*)? Whilst we are unable to offer a specific experiment to address this issue, we feel it is a problem of sufficient magnitude that we are happy to shift the onus to resolve it on to you.*

It is not the case that this data adjustment was used to arrive at our conclusion regarding 4-PBA effects on packaging of ER residents and misfolded proteins. To be clear, our results show that when p24 packaging is inhibited by 4-PBA, COPII vesicles package residents. This striking reciprocal effect—resident levels go up as p24 levels go down—is clear in Figure 3, and we stated in the original text that no data correction is required to reach this conclusion: we wrote “Note that the degree of purification is inherent to the raw data in Figure 3; i.e. it is unaffected by the corrections for changes in budding yield that are made for data presentation in Figure 2, Figure 3”.

Reviewer 2 raises concerns about our treatment of data. While we recognize this is an expert reviewer, we strongly contest this point of view, and we consider it unfortunate that doubt is cast on what in our opinion is an excellent method of controlling for effects in the in vitro budding reaction. We give a brief rebuttal here, with additional comments on this issue in our response to reviewer 2 below.

We will avoid the use of “normalization” in our response, because it can mean different things. What is happening in our budding experiments is very simple: when 4-PBA or point-mutant COPII is used, fewer vesicles form. Reviewer 2 is concerned by this effect, and three questions arise:

1) Is this effect surprising?

2) Does it constitute a “gross defect in vesicular transport” and “dramatic reduction in vesicle formation”?

3) Can it be properly controlled in our experimental approach?

1) The effect is not surprising; in fact it is predicted by the Wieland/Rothman (Cell, 1999) concept of coupling between vesicle budding and p24 protein packaging in the COPI system (explored further by Sandmann and Spang). We make this point in the text, and cite these two studies. This effect was observed for COPI, has not been observed previously for COPII cargo, but should not surprise an expert reviewer.

2) We infer (and state in the text) that the reduction in vesicle budding is based on loss of a bond between the coat and membrane (the essence of the mechanism described by Wieland/Rothman for the dependence of COPI budding magnitude on the p24 interaction). Of course there is no way to know the stage of the budding process that is impacted by the treatment, and we recognize that “loss of a bond” belies the complexity of the budding process. But our structural analysis, together with previous study of B-site mutation in the yeast system, establishes that mutation and 4-PBA binding are exceptionally subtle structural alterations. In sum, there is no gross defect here.

3) We quantify the packaging of not one but four distinct proteins to provide a measure of vesicle budding. These are well-characterized SNARE proteins and cargo receptors that bind to COPII using signals distinct from B-site ligands. We reiterate that this is a simple, straightforward, and rather good approach to provide a metric for vesicle budding to control for changes due to the experimental treatments.

We hope, given this explanation, that Reviewer 3 will recognize that the suggestion of increasing Sec23/24 in our assay would be counterproductive by making it more difficult to compare vesicle abundance among the different treatments.

The Reviewing Editor questions how 4-PBA is able to cause a net increase in export of ERQC clients if COPII budding is inhibited. If we assume that inhibition occurs in cells, then there are two parts to the answer. One is that the forward and retrograde transport steps are tightly coupled between ER and Golgi—this is well established (e.g. Gaynor and Emr, 1997). Secondly, a key parameter here is the capacity of the retrieval system. High capacity relies on the forward transport of the KDEL receptor exceeding forward transport of KDEL ligands (we infer that mutant LDL receptor can be retrieved bound to a KDEL ligand, as per Yamamoto, 2001). Hence, the absolute rate of forward transport is not critical; what matters is that 4-PBA reduces the relevant ratio of forward transport of KDEL receptor versus ligands. This interpretation is made in the text toward the end of section “Extracellular secretion of the KDEL-tagged resident chaperone GRP94”.

*Reviewer #1:*

*This paper reports on the discovery that 4-phenyl butyrate engages a site on the SEC24 subunit of COPII coat normally reserved for cargo receptors and goes on to reveal a surprising equivalence between the defect thereby imposed on cargo incorporation into vesicles in an* in vitro *preparation exposed to 4-PBA and one constituted with COPII coat mutated in the cargo receptor site. Further evidence is provided to support the idea that 4-PBA's ability to encourage the trafficking of misfolded secreted protein (normally retained by ER quality control, ERQC, apparatus) derives from its ability to block cargo incorporation into export vesicles. These findings shed light both on fundamentals of the constitution of ERQC apparatus and on the action of a mysterious if chemically simple compound.*

*Specific comments:*

*Figure 6 correlates the affinity of 4-PBA analogues for the B site of SEC24 with their potency in encouraging trafficking of mutant LDL out of the ER in live cells. An extension of this correlation to the effects of an informative subset of these analogues in the* in vitro *budding assays would provide further confidence in the model (whereby neutering cargo receptor binding to COPII contributes to 4-PBA's effect on client protein trafficking).*

Please see our response above.

*In a similar vein (and perhaps this has already been noted previously): Given that 4-PBA is a low affinity ligand for the B site; has it been shown that higher affinity ligands also work against incorporation of cargo and favor incorporation of ER residents and misfolded protein into vesicles that bud out of the ER in the assays shown in Figure 2–Figure 4? If not, such an experiment should be added in validation.*

To our knowledge, our discovery of 4-PBA target and mechanism of action is novel in the field. We share the reviewer’s interest in higher-affinity 4-PBA analogs. In clinical study (pioneered by Pamela Zeitlin) of cystic fibrosis patients with the F508Δ mutation, 4-PBA showed some modest effects, specifically a partial restoration of CFTR activity in nasal epithelium. However, it is evident that the PK properties of 4-PBA are inadequate: the peak plasma concentration (~1.5 mM) is a small fraction of the IC_50_ for effect in cultured cells, and turnover by β-oxidation is rapid.

*Plotted in 4C is the relative recovery of Cnx/Grp94/ERp57 etc normalized to P24. This indeed showcases nicely the selective effect of 4-PBA and mutant Sec24 on packaging of cargo receptors, but given the overall decline in vesicle budding induced by the chemical, it is hard to understand how its application results in a net increase in flow of resident ER and (presumably misfolded) client proteins. Are there regulatory mechanisms operative* in vivo *that defend total vesicular transport out the ER despite the defect imposed by 4-PBA or the mutant SEC24/COPII?*

Please see our response above in Essential revisions.

*Reviewer #2:*

[…]

*1) The authors elide the somewhat cloudy issue of the precise export signal on p24 proteins, and cite only the Nakamura and Nufer papers that delineate the C-terminal ΦC motif. Belden and Barlowe, however, published a FF/YF motif in the yeast proteins that was shown to interact directly with COPII (both Sec23/24 and Sec13/31, somewhat confusingly). The utility of this signal was then demonstrated* in vivo *by our lab (D'Arcangelo et al. 2015). Our own findings that looked at packing and mutual protein stability suggest that the FF/YF motif is a bona fide export signal, but of course don't rule out that there is an additional contribution from the ΦC motif. It's tempting (for me) to speculate that the two signals have subtly different affinities for the different Sec24 isoforms that might be of relevance here and in the yeast work that suggests ligand-binding alters affinity for different Sec24 isoforms (Muniz lab). So, I appreciate that the authors don't fully believe the Barlowe/Miller lab findings and simply avoid this discussion rather than being dismissive. However, for a reader not intimately familiar with the research history, this omission makes the field more confusing.*

With apologies, we must write that this is not factually correct. Belden and Barlowe, 2001 did not show evidence for specific interaction of a FF/YF motif with Sec23/24. They wrote: “…binding of the Sec23p-Sec24p complex was not affected by any of these mutations when compared with wild-type sequences”. Nakamura et al., 1998 made a direct test of whether the FF motif of the yeast p24 protein Emp24p is an ER export signal. They mutated FF to AA and measured whether this reduced the rate of ER to Golgi transport of their reporter molecule (their construct “WWA”). They observed that the mutation increased the rate of transport. The authors wrote: “…..the mutation [of FF to AA] increased the transport rate… This clearly showed that the double-phenylalanine residues were not the anterograde transport signal.”

The elucidation of the ΦC signal by the Hauri lab and the dissection of the signal on the p24 protein by Nakamura et al. are models of clarity. On the basis of facts gleaned from those studies we made novel predictions regarding 4-PBA mimicry of the ΦC signal and its targeting of COPII that would not be possible based on an alternative set of facts.

*2) In relation to the above point, I note that the peptides used for crystallography only encompass the C-terminal 6 residues, therefore lack the FF/YF motifs described above. In many species, these motifs are flanked by an acidic Glu residue: is it possible that this could be accommodated in the B-site and contribute to the charge interactions that the carboxylate mediates in the ΦC motif?*

Following from 1), there is no logical reason for us to include the FF/YF motifs in our study. Ultimately, the crystallographic results we obtained show recognition of the various ΦC signals by COPII that agrees with Nufer and Nakamura functional data. It is clear from D’Arcangelo, 2015 that mutation of the FF/YF sequence affects p24 protein trafficking, but there is no reliable evidence that these are signals in the conventional sense of polypeptide sequences that bind to coat protein. The reviewer’s suggestion regarding a FF-acidic motif does not conform to known transport signals, and seven out of eight yeast p24 paralogs lack this acidic residue—all eight possess a ΦC motif. For researchers interested in exploring the role of the FF/YF sequence, we would suggest the possibility that this pair of aromatic residues is conserved at the juxta-membrane position to break the stability of the coiled coil of the p24 complex as it emanates from membrane, so as to ensure that the ΦC signal is presented in a flexible context for interaction with the B-site of Sec24.

*3) The authors have previously clearly demonstrated the mechanism of cargo discrimination by Sec24a/b and Sec24c/d isoforms at the B-site. Now they show that p24 proteins and 4-PBA are equal substrates for all isoforms. The mechanism by which this differs from the other peptide interactions at the B-site should be more fully explained/explored.*

We thank the referee for this suggestion. We have included an explanation at the end of the section on “Vesicle packaging of p24 proteins is inhibited by 4-PBA and by a ΦC binding-site mutation”. The specificity of Sec24a/b for LxxLE and DxE signals was established by earlier functional analyses. We gave a structural interpretation in Mancias and Goldberg, 2008 as to why Sec24c/d do not recognize these signals, though never analyzed the details comprehensively, e.g. by mutagenesis. The lack of specificity among Sec24a/b/c/d toward 4-PBA and the ΦC signal is easier to explain, because the contacting residues are almost universally conserved at the B-site of the Sec24 a/b/c/d paralogs.

*4) I'm very uncomfortable with the method of normalization of vesicle budding experiments. Here, I'm afraid I don't have a good alternative suggestion, given the realities of the experiments. I would note, however, that when we and others (Wieland lab) have used mutant Sec24 isoforms in budding experiments, we generally don't see a decrease in vesicle release. This has always been an important indicator that the mutations don't perturb COPII function more broadly but simply report on cargo enrichment. So, I don't fully understand why there is such a dramatic reduction in vesicle formation. I also don't think that normalization for vesicle release by monitoring an "unaffected" cargo protein is the best solution to this problem. Again, in the past, we and others have used lipid release as a measure of vesicle formation.*

Following on from our response in Essential revisions, there are a few additional points we wish to make. First, the Wieland lab was certainly cognizant in their work of the need to measure the production of vesicles, and of the utility of an “unaffected” cargo protein. In Adolf, Rheil, Reckmann and Wieland, 2015 they state: “The cycling cargo adaptor protein ERGIC53, a Sec24 isoform-independent constituent of COPII vesicles (Mancias and Goldberg, 2007), served as a measure for the amount of vesicles generated. Based on the signal intensity of ERGIC53, similar amounts of COPII vesicles were generated when either the Sec23A/24A or Sec23A/24C coat subcomplex was used”. Thus, the procedure in that study and in our own lab (Mancias and Goldberg, 2007) was to use a B-site ligand to control for the effect of mutation at the IxM-binding site, whereas in our present study we use IxM-site ligands to control for the effect of mutation at the B-site. The difference of course is that B-site mutation (and correspondingly 4-PBA) reduces vesicle budding.

The use of lipid release as a measure of vesicle formation is possible with washed yeast microsomal membranes, but is clearly not applicable to the mammalian permeabilized cell budding assay.

*5) Is there an explanation for why ER resident packaging in the 50/50 wt/mutant COPII condition is not equivalent to the 4-PBA treatment? I thought that the point of this condition was to mimic that of the 4-PBA in terms of degree of reduction of p24 packaging. If p24 depletion is the same and vesicle release is equivalent, shouldn't capture of ER residents also go up? It might be a good idea to show the normalization blots for each experiment.*

The explanation is that 10 mM 4-PBA is not perfectly equivalent to the 50/50 mix. 10 mM 4-PBA is higher than the EC_50_ of 6.4 mM as measured in the transport assay; this is mentioned in the section “Vesicle packaging of p24 proteins is inhibited by 4-PBA and by ΦC binding-site mutation”. We can make only a rough estimate of this, but our results altogether suggest that 10 mM 4-PBA is really equivalent to a 40/60 or 35/65 ratio of wt/mutant COPII in the budding assay. For our experiments, it seemed reasonable to us to choose a 50/50 mix for the sake of simplicity.

Although it is a marginal effect, close inspection of Figure 2, Figure 3 and especially 4 suggests that 10 mM 4-PBA causes on average slightly more inhibition of p24 packaging and somewhat more resident packaging than the 50/50 treatment. Note that we do not know the scaling reciprocal relationship between p24 depletion and resident packaging.

*6) I have one suggestion for an experiment to validate the proposed model and substantiate the correlation between 4-PBA effects and Sec24 inhibition: the authors could overexperess Sec24 in cells, both wt and mutant, and repeat the 4-PBA titration and LDL-R/GRP94 leakage assays.*

As stated above in Essential revisions, an ectopic expression experiment cannot give evidence for a mechanism-of-action target. Additionally, to cause just a modest displacement of 4-PBA from a millimolar target, Sec24 protein would have to be expressed in cells at unfeasibly high levels. The ideal experiment in this context would involve a 4-PBA-insensitive separation-of-function mutant Sec24, but not possible since the 4-PBA and ΦC-signal binding sites overlap. We feel we must comment that it is perplexing to us why the referee would suggest the ectopic expression experiment when the potency/affinity experiment summarized in Figure 6 is self-evidently superior, and when we have reported multiple additional tests of the target and mechanism of action of 4-PBA.

*7) Finally, I really like the steric model proposed by the authors as to how p24 proteins might act to exclude ER resident proteins. I note that this concept was also proposed by Chris Kaiser (PNAS 2000) and the steric effects of cargo crowding noted in a review by Stachowiak/Brodsky/Miller (Nature Cell Biol. 2013).*

These words could be taken to imply that we are claiming priority for an idea due to others. The referee may not have intended this; it is certainly not the case. Since our model (described in “Conclusion”) is so speculative, we gave only an outline, but we think it should be clear that we are suggesting not a steric but more a partitioning model in which resident complexes are excluded by the p24 array based on size rather than by competition with cargo. This is closer to the review of Hendershot, 2000, who summarized the results of the Lippincott-Schwartz study (Nehls, 2000), which suggested that resident complexes were small enough to be mobile in the ER but large enough to be excluded from vesicles somehow. Chris Kaiser in his 2000 review emphasized p24 proteins as “placeholders” that cargo molecules would have to displace from the COPII bud. We don’t value our model over the others, and it is worth pointing out that none of the models can explain the very interesting phenomenon that ER residents are packaged into COPII vesicles at an elevated rate in UPR-activated cells.